# The First Swedish Outbreak with VIM-2-Producing *Pseudomonas aeruginosa*, Occurring between 2006 and 2007, Was Probably Due to Contaminated Hospital Sinks

**DOI:** 10.3390/microorganisms11040974

**Published:** 2023-04-08

**Authors:** Carl-Johan Fraenkel, Gustaf Starlander, Eva Tano, Susanne Sütterlin, Åsa Melhus

**Affiliations:** 1Department of Infectious Diseases and Hospital Infection Control, Lund University Hospital, SE-22185 Lund, Sweden; 2Section of Clinical Microbiology, Department of Medical Sciences, Uppsala University, SE-75185 Uppsala, Sweden; 3Department of Women’s and Children’s Health, Uppsala University, SE-75185 Uppsala, Sweden

**Keywords:** *Pseudomonas aeruginosa*, sink, nosocomial outbreak, MBL, VIM-2

## Abstract

Multidrug-resistant *Pseudomonas aeruginosa* is an increasing clinical problem worldwide. The aim of this study was to describe the first outbreak of a Verona integron-borne metallo-ß-lactamase (VIM)-2-producing *P. aeruginosa* strain in Sweden and its expansion in the region. A cluster of multidrug-resistant *P. aeruginosa* appeared at two neighbouring hospitals in 2006. The isolates were characterized by PCR, pulsed-field gel electrophoresis (PFGE), and whole-genome sequencing. Patient charts, laboratory records, and hygiene routines were reviewed, and patients, staff, and the environment were screened. The investigation revealed a clonal outbreak of a VIM-2-producing *P. aeruginosa* strain belonging to the high-risk clonal complex 111, susceptible only to gentamicin and colistin. No direct contact between patients could be established, but most of them had stayed in certain rooms/wards weeks to months apart. Cultures from two sinks yielded growth of the same strain. The outbreak ended when control measures against the sinks were taken, but new cases occurred in a tertiary care hospital in the region. In conclusion, when facing prolonged outbreaks with this bacterium, sinks and other water sources in the hospital environment should be considered. By implementing proactive control measures to limit the bacterial load in sinks, the waterborne transmission of *P. aeruginosa* may be reduced.

## 1. Introduction

*Pseudomonas aeruginosa* is a non-fermenting, Gram-negative rod commonly found in soil and aquatic environments. This opportunist has a high ability to form biofilms and develop antibiotic resistance, and it is one of the leading agents of nosocomial infections and outbreaks. The outbreaks are often localized to units with vulnerable patients and a high consumption of broad-spectrum antibiotics, e.g., intensive care units (ICUs) [1,2,3]. Staff, co-patients, visitors, medical equipment, water supplies, sinks, hygiene products, etc., can all take part in the transmission [1,4,5,6,7,8].

β-lactams with or without β-lactamase inhibitors are often the first-line therapy for severe infections caused by *P. aeruginosa*, but this bacterium may render β-lactams useless through a range of complex resistance mechanisms [9,10,11,12,13]. Several β-lactamases can be carried by *P. aeruginosa*, but the metallo-β-lactamases (MBLs) have, in recent years, become more frequent. MBLs can hydrolyse all β-lactams used for treating *P. aeruginosa*-induced infections except aztreonam, and these enzymes have rapidly become a major concern [14,15]. MBL-encoding genes are typically embedded in integrons and they are transferable by plasmids. On the plasmids, the MBL genes are often combined with resistance genes covering other classes of antibiotics. The result is often multidrug-resistant isolates, leaving physicians with few, if any, therapeutic options [14,15,16].

The most prevalent transferable MBL gene in clinical samples is Verona integron-borne metallo-ß-lactamase (VIM)-2 [11]. It was identified in 1996 in a *P. aeruginosa* isolate from a patient in France [17]. Ever since, the VIM-2 gene has, in the vast majority of cases, been recognized in *P. aeruginosa* isolates, and, as a consequence, it is involved in most outbreaks of MBL-producing *P. aeruginosa* [2,3,18,19,20]. The first VIM-2-producing *P. aeruginosa* in Sweden was isolated in 1999 at Malmö University Hospital in the south of Sweden [21]. The patient died several months later and no new cases were reported until 2004, when two new multidrug-resistant *P. aeruginosa* isolates appeared a few months apart. Both patients had been discharged from an ICU at Lund University Hospital to two different hospitals within the catchment area of Malmö University Hospital. The investigation showed that both isolates carried VIM-2, and they exhibited the same DNA pattern as the isolate from 1999 with an arbitrarily primed polymerase chain reaction (PCR) [21]. The Department of Infectious Control in Lund was contacted. No control measures were taken.

In 2006, a cluster of cases of multidrug-resistant *P. aeruginosa* appeared in the neighbouring Blekinge County. The aim of the study was to describe the first outbreak of VIM-2-producing *P. aeruginosa* in Sweden, as well as the characteristics of the outbreak strain and the patients involved. Furthermore, the most likely source of the outbreak, the infection control measures taken, and the continued clonal expansion in the southern part of Sweden are reported.

## 2. Materials and Methods

### 2.1. Settings and Ethics

Blekinge County is in the south-eastern part of Sweden and has a population of approximately 150.000 inhabitants. It has two hospitals, the secondary-level Karlskrona Hospital (KaH) with 330 beds and the primary-level Karlshamn Hospital (KnH) with 120 beds. At the tertiary health care level, the patients are transferred to the university hospitals in Lund or Malmö. These latter two hospitals are only 25 km apart, and patients are often transported between them. During the first three months of 2006, four patients with multidrug-resistant *P. aeruginosa* were observed at KnH (*n* = 3) and KaH (*n* = 1). An epidemiological investigation was initiated after the third patient. As this study was performed according to the Swedish Infection Protection Act and as a part of an outbreak investigation with a direct impact on public health, no ethical approval was needed. Personal identifiers have been removed in order to ensure confidentiality.

### 2.2. Cases

All patients admitted to KaH and KnH between February 2006 and June 2007 with an infection or carriage of a *P. aeruginosa* isolate resistant to imipenem, ceftazidime and a positive Etest for MBLs were defined as cases.

The first MBL-producing *P. aeruginosa* was isolated from a urine sample obtained from patient 1 on 1 February 2006. The patient had contracted a catheter-associated urinary tract infection (CAUTI) during a stay in the ICU at KnH. From 2 February to 6 March 2006, patients 2, 3, and 4 were reported. Patient 2 had ventilator-associated pneumonia and died within a week in the ICU at KnH. Patient 3 was treated for a CAUTI in a medical ward at KnH. Patient 4 was discovered in a surgical ward at KaH after screening. Only carriage was observed, and the patient died within the follow-up period due to underlying conditions. Five months followed without any new patients, and the outbreak seemed to be at an end. However, from 7 August 2006 to 20 May 2007, four additional patients were registered. Patient 5 was admitted to the ICU at KnH with a pneumothorax. After transfer to a medical ward, an MBL-producing *P. aeruginosa* isolate grew in a sputum sample. The patient was transported to the Department of Infectious Diseases at KaH but died within two weeks. Patient 6 shared the room with patient 5 at the medical ward at KnH, and the MBL-producing *P. aeruginosa* was isolated from a bronchial aspirate and a leg ulcer. The patient exhibited no signs of infection. Patient 7 had a urinary catheter and was discovered through screening prior to surgery. Patient 8 received a urinary catheter after surgery and MBL-producing *P. aeruginosa* isolates were found at multiple locations after screening. The patient was transferred to the Department of Infectious Diseases at KaH for isolation, but died within 11 days. A summary of the patient data is shown in Table 1.

### 2.3. Epidemiological Investigation

The patient charts were reviewed, and infections, underlying conditions, antibiotic treatments, and international travels were registered. In addition, the dates for admissions, transfers, and discharges in the last 3 months prior to culture positivity were extracted, and the room numbers in different wards were noted when possible. The laboratory records were used to find culture data concerning *P. aeruginosa* with and without MBL, and to follow how long the outbreak patients stayed culture-positive. Policies and procedures were thoroughly controlled, especially concerning hand hygiene, cleaning, and medical equipment.

Patients with wounds and urinary catheters were screened for the presence of MBL-producing *P. aeruginosa* at medical ward B, KnH, when the bacterium was first isolated from patient 3. Screening was also performed on patients at medical ward A, KnH, and on the staff of the ICU in the same hospital when patients 5 and 6 were diagnosed. Finally, screening of patients alone was performed at the surgery wards at KaH and KnH when patients 7 and 8 were found to be positive. For these patients, rectal samples were added.

From August 2006 to June 2007, 124 environmental samples were collected from contact surfaces and sinks in rooms in the ICU, medical ward A, and the surgery ward at KnH, and in surgery ward A at KaH.

### 2.4. Cultures and Susceptibility Testing

*P. aeruginosa* was identified with conventional laboratory methods and/or an API 20 NE^®^ instrument (BioMerieux, Lyon, France). Antimicrobial susceptibility was tested with disk diffusion on Iso-Sensitest Agar^®^ (Oxoid Ltd., London, UK). Breakpoints established by the Swedish Reference Group for Antibiotics was used (nowadays, exchanged for the European Committee on Antimicrobial Susceptibility Testing).

The MBL phenotype was detected with the Etest for MBLs (AB Biodisk, Solna, Sweden). A reduction in imipenem minimum inhibitory concentrations (MICs) by ≥3 twofold dilutions in the presence of ethylenediaminetetraacetic acid (EDTA) was interpreted as being positive for MBL production. An additional double-disk test with imipenem ± EDTA and ceftazidime ± 2-mercaptopropionic acid (MPA) was performed as earlier described [22,23]. The MICs of MBL-positive strains were further determined for imipenem, meropenem, piperacillin/tazobactam, ceftazidime, aztreonam, gentamicin, amikacin, tobramycin, ciprofloxacin, fosfomycin, and colistin.

### 2.5. Identification of MBL with PCR

Deoxyribonucleic acid (DNA) was prepared by heating a bacterial suspension to 95 °C for 10 min. The template was added to the HotStarTaq master mix (Qiagen AB ^®^, Solna, Sweden) with earlier described primers [24] (Eurogentech S.A., Seraing, Belgium) in a final volume of 25 μL. The PCR reactions were processed in a GeneAmp PCR System 9700 cycler (PE Applied Biosystems, Foster City, CA, USA), where the program was carried out at 94 °C for 5 min, and was followed by 30 cycles of 1 min at 94 °C, 1 min primer annealing at 55 °C, 1.5 min at 72 °C, and a final extension step at 72 °C for 5 min. The PCR products were separated by electrophoresis on 1.5% agarose gels (GeneChoice Inc., Frederick, MD, USA). The patient isolate from 1999 was used as the positive control.

### 2.6. Pulsed-Field Gel Electrophoresis (PFGE)

PFGE was performed as previously described [25]. Banding patterns were compared visually with BioNumericsSoftware, version 4.0 (Applied Maths Bvba, St.-Martens-Latem, Belgium). Using clustering of a similarity matrix based on band-matching Dice coefficients (tolerance 1% and optimization 1%), dendrograms were created. Isolates showing indistinguishable pulsed-field patterns or closely related band patterns (>90% similarity) were regarded as clonally related.

### 2.7. Whole-Genome Sequencing (WGS) and Computational Analysis

One of the VIM-2-positive isolates (from patient 3) was randomly chosen for WGS. One colony was incubated overnight in Luria–Bertani broth (Becton Dickinson, Stockholm, Sweden) at 35 °C. DNA was prepared using the BioRobot M48 system (Qiagen GmbH, Hilden, Germany) and the MagAttract^®^ DNA Mini M48 Kit according to the manufacturer’s recommendations. The DNA preparations were transported to SciLifeLab, Uppsala, Sweden. After controlling the DNA quality, sequencing was carried out on an IonTorrent™ PGM instrument (Life Technologies, Carlsbad, CA, USA) with a read length of 400 bp. Read quality was assessed using FastQC software (v0.11.4, http://www. bioinformatics.babraham.ac.uk, accessed on 1 November 2016), according to the developers’ recommendations.

The obtained read was de novo assembled with the methylated-CpG island recovery assay (MIRA) 4.9.5_2 [26]. The assembly statistics were assessed with the Quality Assessment Tool for Genome Assemblies (QUAST) v4.4 [27], and species confirmation was performed with the ribosomal multilocus sequence typing (rMLST) speciation tool at pubmlst.org/rmlst. Seven-gene MLST was carried out [28] using the database hosted on www.pubmlst.org. The IntegronFinder tool [29] was used for the detection of integron loci in the draft genome, and the predicted integrons were illustrated with the statistical software R (v3.3.3, R Foundation for Statistical Computing, Vienna, Austria, www.R-project.org, package genoPlotR). In addition, the isolate was analysed concerning acquired genetic resistance determinants with the Antibiotic Resistance Gene-ANNOTation (ARG-ANNOT) database [30]. For virulence factors, the Virulence Factors of Pathogenic Bacteria database (VFDB, 2016) [31] was applied together with Basic Local Alignment Search Tool (BLAST) searches on the draft genome [32]. Read sequence data are available from the European Nucleotide Archive, Sequence Read Archive, and DNA DataBank of Japan databases under the project reference PRJEB25448.

## 3. Results

### 3.1. Bacterial Findings

From February 2006 to June 2007, there were a total of 816 clinical *P. aeruginosa* isolates from 507 patients. Of these isolates, 69 (8.5%) were resistant to imipenem and 9 (1.1%) were resistant to both imipenem and ceftazidime. The MBL phenotype was confirmed in all but one of these nine isolates. It was excluded since it did not fulfil the case definition. The remaining eight isolates were all PCR-positive for VIM-2 and had band patterns that were identical or exhibited >90% similarity in the PFGE (Figure 1). They were all susceptible to gentamicin (MIC 2 mg/mL) and colistin (MIC 1 mg/mL). The MICs for the remaining tested antibiotic drugs were the following: carbapenems and ciprofloxacin > 32 mg/mL; fosfomycin, 64 mg/mL; piperacillin-tazobactam, 32 mg/mL; and ceftazidime, aztreonam, tobramycin, and amikacin, 16 mg/mL.

### 3.2. Genetic Data of the Outbreak Strain

A total of 1.2 million sequences with read lengths varying from 8 to 546 bases were obtained. The reads were de novo assembled into a draft genome, resulting in a total assembly length of 7.2 million bp; it was 121 contigs larger than 100,000 bp with an average coverage per contig ranging from 26 to 51, and an N50 of 69,517. The guanine–cytosine (GC) content of the draft genome was 65.73%, and 96.6% of the draft genome could be mapped to the complete genome of *P. aeruginosa* PA01 (NC_002516.2).

Sequence typing revealed that the isolate belonged to the clonal complex 111. The assembly failed to produce the *acs* locus, which is why the exact sequence type could not be determined. The remaining alleles were: *gua*—allele 5; *mut*—allele 5; *nuo*—allele 4; *pps*—allele 4; and *trp*—allele 3.

IntegronFinder predicted one complete integron class I with the *bla_VIM_*_-2_ and an uncharacterized protein (Figure 2). The integron consisted of the integrase gene (*intI*), two promotors Pc and P_int_, and the *attI* recombination site, followed by the gene cassette with an open reading frame (ORF) of 218 bp, the attC recombination site, and the beta-lactamase *bla_VIM_*_-2_. Additionally, two clusters of attC sites lacking integron integrases (CALIN) were detected on separated contigs, both with two ORFs with uncharacterized proteins.

The BLAST search in the ARG-ANNOT database on the draft genome yielded the following resistance determinants: two beta-lactamase genes *bla*_VIM-2_ and *bla*_OXA-50_, two aminoglycoside resistance genes *aph3*-*IIb* (O-phosphotransferase) and *aacA29b* (N-acetyltransferase), the chloramphenicol acetyltransferase *catB7* (a chromosome-encoded variant of the cat gene found in *P. aeruginosa*), and the fluoroquinolone resistance gene *oqxB*. The *oprD* gene included frameshifts.

The BLAST search for virulence factors confirmed the existence of all chromosomal loci related to the pathogenicity of *P. aeruginosa* PA01 in the VFDB, including the HIS-I Hcp1 secretion island I and TTSS type III secretion system.

### 3.3. Epidemiological Investigation

All wards visited by the case patients during the 3-month period prior to culture positivity are listed in Table 1. Wards and rooms shared by patients in relation to time are shown in Figure 3. None of the patients had travelled outside Scandinavia prior to their culture positivity, but two patients (patient 2 and patient 4) had received treatment at Lund University Hospital during 2003.

All patients but one (patient 4) were first admitted to KnH. Patients 1–3 had all stayed in the medical ward B at KnH, but not at the same time (see Figure 3 for the time interval between the patients). Data are missing concerning which room they stayed in after admission to this ward, but it is unlikely that the outbreak strain was transmitted to patient 3 in this ward since the patient was already culture-positive on the day of admittance. Patients 1, 5, 6, and 8 had a period of care in medical ward A at KnH. Patients 5 and 6 had stayed in the same room, whereas patient 1 had been admitted to the ward about 2 months earlier and patient 8 about 9 months later. Patients 1, 2, 5, and 7 had, in addition, stayed in the same room in the ICU at KnH, but not at the same time (Figure 3). At KaH, patients 4 and 7 had been admitted to ward A in the Department of Surgery, but their admission was separated by 9 months.

Of the 124 environmental samples, MBL-producing *P. aeruginosa* was found in 2. According to PFGE, both isolates belonged to the outbreak strain (not shown). One of the positive environmental isolates was collected on 15 August 2006 from a sink drain in room 2 at the ICU, KnH, where patients 1, 2, and 5 had stayed 34, 26, and 2 weeks earlier, respectively. The second environmental sample was obtained on 24 May 2007 from the sink drain in a room at the surgical ward, KnH, when patient 8 was still there. In no instance was the outbreak strain found among the screened staff.

### 3.4. Infection Control Measures

Once infection/colonization was established, all case patients were isolated in single rooms with separate toilets and showers. Healthcare workers were instructed to use gowns and gloves when in contact with the patients. Alcohol-based hand rub was used before and after glove use and patient contact. From July 2006, case patients were transferred to and isolated at the Department of Infectious Diseases, KaH. The ICU at KnH was closed for admission for one week in August 2006 when the outbreak strain was discovered in one of the sink drains. This sink was replaced, the other sinks were treated overnight with acetic acid (24%), and the ward was disinfected with Virkon^®^ (Viroderm, Solna, Sweden). No MBL-producing bacteria could thereafter be isolated from any of the sink drains at the ICU. The sink in the surgical ward at KnH was treated with boiling water and 24% acetic acid. Cultures obtained from the sink one week later were negative for MBL-producing *P. aeruginosa.* During the following 15 years, no more cases were reported.

## 4. Discussion

In the present study, the first Swedish outbreak involving MBL-producing *P. aeruginosa* was described. The outbreak was prolonged, and the causal strain was multidrug-resistant, VIM-2-producing, and belonged to the high-risk clonal complex 111. During a period of about 1.5 years, eight patients were colonized and/or infected with the outbreak strain in Blekinge County. Fifty percent of them developed a clinical infection and three died.

Due to the separation in time, patient-to-patient transmission was unlikely in all cases but one. However, several patients had been admitted to the same rooms or wards for shorter or longer time intervals, which is why a source in the environment was most likely. The environmental samples yielded growth of the outbreak strain in two sinks located in rooms where the colonized patients had stayed, providing a likely reservoir. As the sinks were replaced or decontaminated, no more cases occurred. It has remained so during the last 15 years in Blekinge County.

The isolation frequency of VIM-producing *P. aeruginosa* has been very low in Sweden and other Scandinavian countries. Most isolates have been derived from patients recently hospitalized abroad, suggesting that the import of international clones has been the major route for acquiring this type of bacterium [21]. The only exception is the southern part of Sweden, where the local expansion of ST111 has caused several infections and deaths. Although the intervention was successful in Blekinge County, this clone continued to cause infections. In 2007, four new patients appeared at Lund University Hospital [21]. They were followed by twelve more cases during the years between 2008 and 2013 [33]. Due to the low prevalence, these infections are difficult to recognize and easy to ignore. However, the mortality is high, and in Lund it was 50% [33]. To stop the transmission of VIM-2-producing *P. aeruginosa*, 24% acetic acid was used and a proactive routine was introduced to reduce the bacterial load; since 2013, the sinks in high-risk wards have been treated with acetic acid. At the beginning it was once weekly, but in later years the interval has been prolonged to once every month or every three months. No more clusters or outbreaks have been recorded.

It is obviously easier to stop an outbreak of *P. aeruginosa* than to eradicate this bacterium from a plumbing system. The fact that the outbreak strain has resisted ten years of acetic acid treatment emphasizes the importance of acting promptly to avoid the establishment of biofilm further down the drain system. This finding also has future implications, as multidrug-resistant bacteria are expected to become more common. Outbreaks may be on a large scale and cost a lot, but they do not cost as much as constantly present healthcare-associated infections (HAIs). These everyday infections are also more difficult to discover unless the bacteria carry some marker in the form of resistance genes. Recently, a British research group showed that environmental, Gram-negative bacterial populations are largely structured by ward and sink, with a few lineages being widely distributed [34]. They also compared the environmental isolates with contemporaneous patient isolates and reported that sinks may contribute to up to 10% of the infections caused by *E. coli* [34]. In a small country such as Sweden, HAIs affect approximately 65.000 patients every year at a cost of EUR 150–220 million. About 40% of these infections are urinary tract infections, and a majority of them are caused by *E. coli*. Several million EUR would probably be saved yearly, not to mention all the unnecessary suffering and deaths that could be avoided, if routines regarding sink practices were improved.

Despite extensive investigation, it was never clarified when or where patient 1 acquired the VIM-2-producing *P. aeruginosa* or if this patient was the true index patient in Blekinge County. Patients 2 and 4 had been treated at Lund University Hospital three years earlier, but no other connections with the university hospitals in Lund or Malmö were established. It is possible that ST111 could have been introduced repeatedly into this region, but it seems more likely that the clone was present in different sinks for all these years, and that Lund University Hospital played a central role in delivering new patients.

It was also not easy to follow the transmission routes for the patients at KnH. Some of the patients had been admitted to the same rooms or wards, but they were separated in time by weeks or months. Direct contact was unlikely, except for the two patients who had stayed in the same room at the same time. None of the staff were a carrier; thus, it was most likely that patients were colonized from a reservoir in the environment. In support of this hypothesis were the findings in the two sinks, of which one was in room 2 at the ICU where half of the patients had stayed. *P. aeruginosa* survives for only shorter time periods on dry surfaces, but anything wet or moist in an environment may act as a reservoir [35,36]. The bacterium has been associated with water taps, sinks, plumbing systems, shower drains, and faucets in hospitals. To make the outbreaks cease, replacements of the water fittings, sinks, etc., are often necessary [1,8,37,38,39]. In some hospitals the solution has been more drastic, and waterless patient care has been implemented [40].

The outbreak strain belonged to the high-risk multi-drug resistant CC111, which has been one of the most successful clonal complexes globally [3,39,41,42]. It is known for its virulence and ability to produce biofilms. Apart from *bla*_VIM-2_, which was linked to, but not within, the class 1 integron, it had frameshifts in the *oprD* gene and carried *bla*_OXA-50_. This relatively weak β-lactamase is constitutively expressed in *P aeruginosa* and confers decreased susceptibility to piperacillin, and, interestingly, to meropenem in *P. aeruginosa*, but not in *E. coli* [43]. In addition, it was resistant to the most common treatment alternatives, thereby affecting the outcomes for two patients.

In conclusion, the first outbreak of VIM-2-producing *P. aeruginosa* in Sweden had low prevalence and was caused by the high-risk CC111. It lasted 1.5 years and involved eight patients, of whom half developed an infection. Sinks in the rooms were the most likely source, and sinks continued to play a major role as a source during the following years in this part of Sweden. When facing prolonged outbreaks with this bacterium, sinks and their drain systems should be considered. By implementing proactive control measures to limit the bacterial load in sinks and water fittings, the transmission of *P. aeruginosa* to vulnerable patients could probably be reduced, if not eliminated.

## Figures and Tables

**Figure 1 microorganisms-11-00974-f001:**
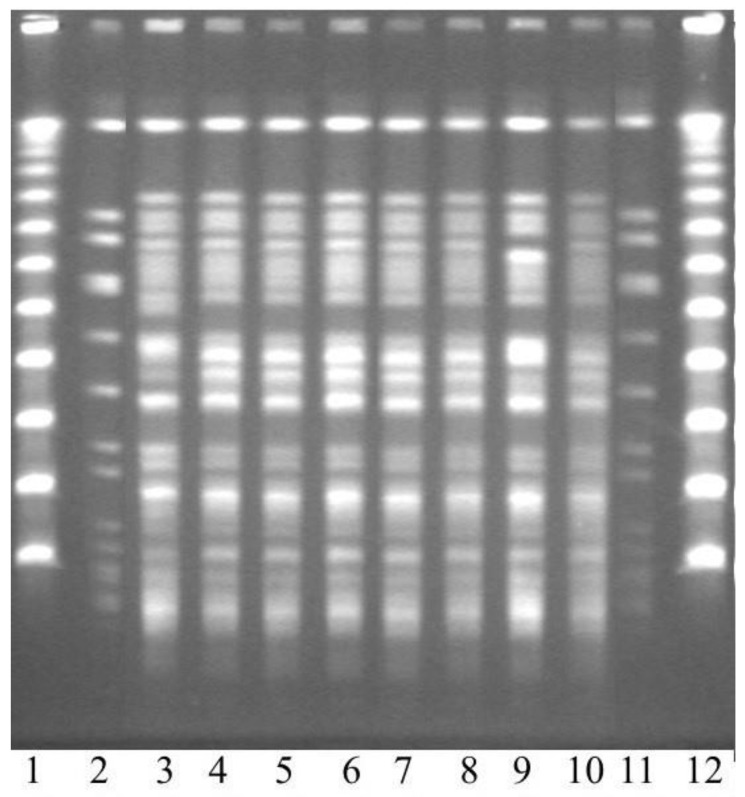
Pulsed-field gel electrophoresis of the patient isolates. Lanes 1 and 12: DNA size markers; lanes 2 and 11: a control strain; and lanes 3–10: patient isolates from the outbreak in the same order as they were isolated.

**Figure 2 microorganisms-11-00974-f002:**
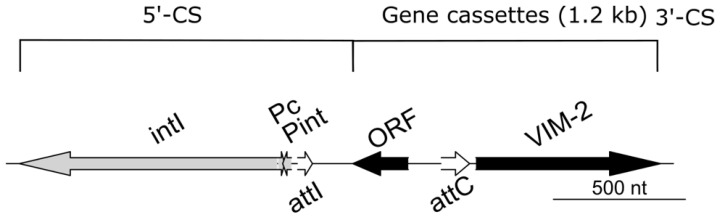
Illustration of the class I integron (2518 bp) of the outbreak strain. Abbreviations: *intl—*class integrase gene; P_c_ and P_int_—gene cassette promotors; *attI—*integron-associated recombination site; ORF—open reading frame; *attC—*recombination site of the gene cassette; VIM-2—Verona integron-borne metallo-ß-lactamase-2.

**Figure 3 microorganisms-11-00974-f003:**
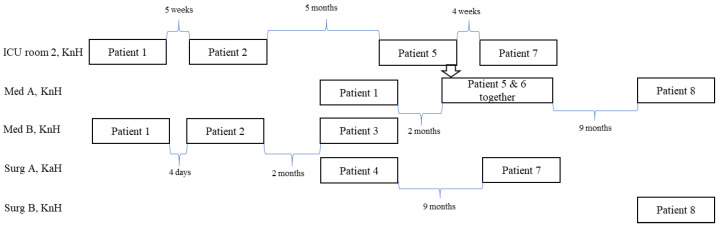
Wards visited by the patients involved in the outbreak. Med A and B—wards in the Department of Medicine, Surg A and B—ward in the Department of Surgery, ICU—intensive care unit, KnH—Karlshamn hospital, KaH—Karlskrona Hospital.

**Table 1 microorganisms-11-00974-t001:** Summary of the patient data.

Patient	MBL Culture + Sample	MBL Screening Prior to Admittance MBL-Induced Infection	Antibiotic Treatment ^1^	Risk Factors	Outcome	Follow-Up	Possible Transmission of MBL+ *P. aeruginosa* ^2^
1	2 February 2006 Urine	Not performed Yes	GEN + CXM	Urinary catheter	Discharged on day 117	MBL− on days 15, 21, 33, and 221	KnH: Med A, Med B, and ICU
2	15 February, 2006 BAL ^3^	Not performed Yes	GEN + PTZ	Invasive ventilation	Dead on day 7	-	KnH: Med B and ICU
3	28 February 2006 Urine	Negative Yes	PTZ	Urinary catheter	Discharged on day 9	MBL− on days 9, 111, and 154	KnH: Med B
4	6 March 2006 Urine	Not performed No	-	Urinary catheter	Discharged on day 4, dead of other reasons	-	KaH: Surg A
5	7 August 2006 Sputum	Negative Probable	AZI, ERT, RIM	-	Dead on day 33	-	KnH: Med A
6	12 August 2006 Sputum, wound	Negative No	CTX	Leg ulcer	Discharged on day 26	MBL− on days 46, 89, and 300	KnH: Med A and ICU
7	13 December 2006 Urine	Negative No	-	Urinary catheter	Discharged on day 2	MBL+ on day 320	KnH: Med A
8	20 May 2007 Urine	Negative No	AMX	Urinary catheter	Dead on day 11	-	KnH: Med A KnH: Surg B

^1^ Amoxicillin—AMX, azithromycin—AZI, ciprofloxacin—CIP, clindamycin—CLI, cefoxitin—CTX, cefuroxime—CXM, doxycycline—DOX, ertapenem—ERT, erythromycin—ERY, imipenem—IMI, metronidazole—MTZ, norfloxacin—NOR, piperacillin-tazobactam—PTZ, rifampicin—RIM, sulfamethoxazole-trimetroprim—TSU. ^2^ Ward and hospital in which a possible transmission of the MBL-producing outbreak strain could have occurred between patients. Med ward—ward in the Department of Medicine, Surg ward—ward in the Department of Surgery, ICU—intensive care unit, Inf—the Department of Infectious Diseases, Ort—the Department of Orthopaedic Surgery, Rehab—the Department of Rehabilitation, Thx—the Department of Thoracic Surgery, KnH—Karlshamn Hospital, KaH—Karlskrona Hospital. ^3^ Bronchoalveolar lavage.

## Data Availability

More detailed patient data are not openly available to ensure confidentiality, but are available from the corresponding author upon reasonable request.

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
