# Peer review of "The First Swedish Outbreak with VIM-2-Producing Pseudomonas aeruginosa, Occurring between 2006 and 2007, Was Probably Due to Contaminated Hospital Sinks"

_microorganisms, 2023, doi:10.3390/microorganisms11040974_

Round 1
Reviewer 1 Report
Report
The present research microorganisms--2268774-peer-review-v1 titled: “The first Swedish outbreak with VIM-2-producing Pseudomonas aeruginosa was prolonged and probably due to contaminated hospital sinks” was aimed at evaluating the first outbreak of a VIM-2-producing MDR P. aeruginosa strain in 11 Sweden and its expansion in a region at two neighboring hospitals in 2006. The topic is very interesting from the medical and environmental aspect, however; there are some major and minor comments and suggestions that should be considered and fulfilled, and these are as follows:
Major Comments
1. In the Introduction, L39-40, it was mentioned that “MBLs can hydrolyse all β-lactams used for treating P. aeruginosa-induced infections, and these enzymes have rapidly become a major concern [14, 15].” Should be changed to “MBLs can hydrolyse most β-lactams, except for aztreonam used for treating P. aeruginosa-induced infections, and these enzymes have rapidly become a major concern [14, 15, Elshamy and Aboshanab, 2020 ]. Therefore, I advised this change to this sentence and include this relevant citation as well Elshamy AA, Aboshanab KM. A review on bacterial resistance to carbapenems: epidemiology, detection and treatment options. Future Sci OA. 2020 Jan 27;6(3):FSO438. doi: 10.2144/from- PMID: 32140243; PMCID: PMC7050608. 2. Abbreviations should be firstly described at the first mention and then used consistently in the whole manuscript (examples, in the abstract, L11, Veronese imipenemase (VIM) and in the introduction at the first mention should be also included (L45). I advised the authors to reeeee the whole manuscript for this matter. 3. L45, In the Methods, the author mentioned that “no ethical approval was needed” However, I advised and recommend having hospital approval since the data of this study are obtained from hospital records particularly, the clinical cases and causes of patient mortality enrolled in this study. However, since there was no direct contact with patients and unidentified patients, this study is eligible for waiving the patient consent, but hospital approval should be included in order to publish such hospital records. 4. L98, “Dep.” Should be changed to “department. 5. In table 1, the abbreviation of BAL should be included in the table footnote. Also, P. aeruginosa should be italicized. 6. In Methods, L160, the author should explain or give the rationale why they selected the isolate from patient 3 for the full genome sequence and not did that for the 8 isolates recovered from this study. 7. In the results, L185, the authors mentioned that” The MBL phenotype was confirmed in all but one of these nine isolates” it seems to be an incomplete sentence so, I advise the author to rephrase it for further clarification. 8. In the results, L188-191. The susceptibility pattern revealed that the eight isolates were all positive for VIM-2 and were all susceptible to gentamicin (MIC 2 mg/mL) and resistant to amikacin (16 mg/mL). I wonder about these findings since both antibiotics are of the same class of aminoglycosides amikacin is usually more effective since it lacks the target hydroxyl group in the aglycone unit rendering it to escape-resistant to phosphotransferases. So, when isolates are amikacin sensitive, they should be gentamicin sensitive, however, the vice versa is not true. Therefore, I recommend the authors revise the hospital records or at least give the rationale and scientific bases for these obtained results in the discussion section. 9. In figure 1, I recommend the authors include the size and location (start-end) of the DNA segment that contains the class 1 integron presented in figure 1 in order to facilitate reviewing it on the genome submission of the respective isolate. In addition, all abbreviations should be added in the figure footnote for more clarification. 10. The author should define the amino acid difference between VIM1 and VIM2 to clarify and confirm the presence of VIM2 and not VIM1 (amino acid alignment for example). Since β-Lactamases VIM-1 and VIM-2 have identical amino acid residues that may be involved near or in the active site of these enzymes (Docquier et al. 2003). So, We need to confirm the amino acid sequence of VIM2 in this study. Docquier, J. D., J. Lamotte-Brasseur, M. Galleni, G. Amicosante, J. M. Frere, and G. M. Rossolini.2003. On functional and structural heterogeneity of VIM-type metallo-β-lactamases. J. Antimicrob. Chemother.51:257-266. 11. The outbreak of this study occurred in the period from February 2006 to June 2007 which is more than 15 years ago. I need the author to give the rationale for becoming late in the publication of these findings. And this has to be reflected in the manuscript title for better clarification for readers as it is an old outbreak in Sweden that is not currently identified. 12. The discussion section is well-written and reflects the major outcome of the conducted research.
Therefore, and for the above-mentioned remarks, I advised a major revision of the respective manuscript in its current state taking into consideration the above comments and recommendations before being considered for publication

Author Response
Reviewer 1
Reviewer’s comment 1: In the Introduction, L39-40, it was mentioned that “MBLs can hydrolyse all β-lactams used for treating P. aeruginosa-induced infections, and these enzymes have rapidly become a major concern [14, 15].” Should be changed to “MBLs can hydrolyse most β-lactams, except for aztreonam used for treating P. aeruginosa-induced infections, and these enzymes have rapidly become a major concern [14, 15, Elshamy and Aboshanab, 2020 ]. Therefore, I advised this change to this sentence and include this relevant citation as well Elshamy AA, Aboshanab KM. A review on bacterial resistance to carbapenems: epidemiology, detection and treatment options. Future Sci OA. 2020 Jan 27;6(3):FSO438. doi: 10.2144/from- PMID: 32140243; PMCID: PMC7050608.
Authors’ response: This is a cultural difference. In contrast to CLSI, EUCAST does not have an S category for P. aeruginosa and aztreonam. The whole wild type population is classified as I. In the recommendations from the Swedish Reference Group for Antibiotics, this antibiotic drug is only recommended for P. aeruginosa when treating cystic fibrosis patients with inhalations. Aztreonam is therefore not routinely tested and seldom used when P. aeruginosa is the causative agent. To adapt to a more international view, this sentence has been changed and the reference included.
Reviewer’s comment 2: Abbreviations should be firstly described at the first mention and then used consistently in the whole manuscript (examples, in the abstract, L11, Veronese imipenemase (VIM) and in the introduction at the first mention should be also included (L45). I advised the authors to reeeee the whole manuscript for this matter.
Authors’ response: We could not find any list of accepted abbreviations in the author instructions, something many journals provide. It is therefore unclear when a common abbreviation is accepted and when it is not. We have, however, gone through the manuscript and revised it as advised. We did not use the words “Veronese imipenemase”, since VIM was short for Verona Integron-borne Metallo-ß-lactamase in the original article.
Reviewer’s comment 3: L45, In the Methods, the author mentioned that “no ethical approval was needed” However, I advised and recommend having hospital approval since the data of this study are obtained from hospital records particularly, the clinical cases and causes of patient mortality enrolled in this study. However, since there was no direct contact with patients and unidentified patients, this study is eligible for waiving the patient consent, but hospital approval should be included in order to publish such hospital records.
Authors’ response: Once again there are cultural differences. To get an approval from a hospital is not possible since there is no such decision-making body at any Swedish hospital. Access to Swedish hospital records for research are regulated by the Ethical Review Act, but it is overridden by the Infection Protection Act when the public health is deemed threatened. This was the case with the present study and this is also stated in the manuscript. To protect the patients, sex, age and most details concerning illnesses and underlying conditions have been removed to make identification impossible. If a patient is impossible to identify, only records and bacteria are studied, and the patients are dead, the Ethical Review Act is not applicable.
Reviewer’s comment 4: L98, “Dep.” Should be changed to “department.
Authors’ response: Dept. is changed to Department in all places.
Reviewer’s comment 5: In table 1, the abbreviation of BAL should be included in the table footnote. Also, P. aeruginosa should be italicized.
Authors’ response: The changes are made as suggested.
Reviewer’s comment 6: In Methods, L160, the author should explain or give the rationale why they selected the isolate from patient 3 for the full genome sequence and not did that for the 8 isolates recovered from this study.
Authors’ response: When the outbreak took place, WGS was extremely expensive (about 15,000 EUR/isolate) and used only by a few. When the isolate from patient 3 was finally sequenced, the outbreak in Blekinge was long over. The reason why this randomly chosen isolate was sequenced at all, was to clarify its relationship to the outbreak strain in Lund. Since the WGS result would have no impact on any of the outbreak outcomes, the financial support was a minimum. WGS is definitely better at discriminating between isolates than PFGE, but that does not mean that the old reference method is out of use or cannot identify an outbreak. We agree that it would have been preferable to have all isolates sequenced, but it is impossible for us to sequence them within 10 days, the time given for revision. We have, however, included a photo of the PFGE and added in the text that the isolate from patient 3 was randomly chosen. See also our response to comment 11.
Reviewer’s comment 7: In the results, L185, the authors mentioned that” The MBL phenotype was confirmed in all but one of these nine isolates” it seems to be an incomplete sentence so, I advise the author to rephrase it for further clarification.
Authors’ response: A sentence has been added about the exclusion of the isolate that did not fulfill the case definition.
Reviewer’s comment 8: In the results, L188-191. The susceptibility pattern revealed that the eight isolates were all positive for VIM-2 and were all susceptible to gentamicin (MIC 2 mg/mL) and resistant to amikacin (16 mg/mL). I wonder about these findings since both antibiotics are of the same class of aminoglycosides amikacin is usually more effective since it lacks the target hydroxyl group in the aglycone unit rendering it to escape-resistant to phosphotransferases. So, when isolates are amikacin sensitive, they should be gentamicin sensitive, however, the vice versa is not true. Therefore, I recommend the authors revise the hospital records or at least give the rationale and scientific bases for these obtained results in the discussion section.
Authors’ response: We are not quite sure that we understand this comment. It is a well-known phenomenon that amikacin resistance can be missed when gentamicin is used as the representative for the whole class of aminoglycosides. Amikacin is derived from kanamycin, which gentamicin is not, and they are usually grouped into two different groups within the aminoglycoside class. Consequently, there are differences when it comes to enzyme inactivation. The whole genome sequenced isolate did only contain two genes that encoded aminoglycoside resistance: aph3-IIb (O-phosphotransferase) and aacA29b (N-acetyltransferase). The first enzyme prefer kanamycin as substrate and does not make much of a difference for the two aminoglycosides mentioned above, whereas the second enzyme inactivates amikacin but not gentamicin (Magnet S, Smith T-A, Zheng R, Nordmann P, Blanchard JS. Aminoglycoside Resistance Resulting from Tight Drug Binding to an Altered Aminoglycoside Acetyltransferase. Antimicrob Agents Chemother. 2003; 47(5): 1577–1583. doi: 10.1128/AAC.47.5.1577-1583.2003). Our result is in accordance with the genetics, literature and the data presented by the reviewer, if we have interpreted it correctly. We will therefore leave this comment without any further response.
Reviewer’s comment 9: In figure 1, I recommend the authors include the size and location (start-end) of the DNA segment that contains the class 1 integron presented in figure 1 in order to facilitate reviewing it on the genome submission of the respective isolate. In addition, all abbreviations should be added in the figure footnote for more clarification.
Authors’ response: The size is indicated by the bar on the right bottom side of the figure, but we have also added the exact size in the footnote. The figure has been changed so that the start-end is shown. The exact location of the integron has not been given, since there are indications that it is situated on a plasmid. Finally, the abbreviations have been added to the footnote.
Reviewer’s comment 10: The author should define the amino acid difference between VIM1 and VIM2 to clarify and confirm the presence of VIM2 and not VIM1 (amino acid alignment for example). Since β-Lactamases VIM-1 and VIM-2 have identical amino acid residues that may be involved near or in the active site of these enzymes (Docquier et al. 2003). So, We need to confirm the amino acid sequence of VIM2 in this study. Docquier, J. D., J. Lamotte-Brasseur, M. Galleni, G. Amicosante, J. M. Frere, and G. M. Rossolini.2003. On functional and structural heterogeneity of VIM-type metallo-β-lactamases. J. Antimicrob. Chemother.51:257-266.
Authors’ response: VIM-1 And VIM-2 belong to two different branches of the VIM-family, and they differ genetically with as much as about 10%. The difference in amino acid sequence between the two enzymes is consequently about the same, as shown by the alignment below. We have marked the substitutions between the references for VIM-1/VIM-2 and the beta-lactamase of the study isolate in green. The amino acid sequence of the study isolate corresponds with the VIM-2 reference. The presence of VIM-2 is thereby confirmed.
The amino acid sequence alignment for VIM-1 (NCBI nucleotide accession ACL31202) and the study isolate’s VIM-2.
Query: ACL31202.1 VIM-1 [Pseudomonas aeruginosa] Query ID: lcl|Query_36829 Length: 266
>Study_isolate
Sequence ID: Query_36831 Length: 342
Score:446 bits(1146), Expect:7e-164,
Method:Compositional matrix adjust.,
Identities:217/240(90%), Positives:230/240(95%), Gaps:0/240(0%)
Query 1 MLKVISSLLVYMTASVMAVASPLAHSGEPSGEYPTVNEIPVGEVRLYQIADGVWSHIATQ 60
M K++S LLVY+TAS+MA+ASPLA S + SGEYPTV+EIPVGEVRLYQIADGVWSHIATQ
Sbjct 1 MFKLLSKLLVYLTASIMAIASPLAFSVDSSGEYPTVSEIPVGEVRLYQIADGVWSHIATQ 60
Query 61 SFDGAVYPSNGLIVRDGDELLLIDTAWGAKNTAALLAEIEKQIGLPVTRAVSTHFHDDRV 120
SFDGAVYPSNGLIVRDGDELLLIDTAWGAKNTAALLAEIEKQIGLPVTRAVSTHFHDDRV
Sbjct 61 SFDGAVYPSNGLIVRDGDELLLIDTAWGAKNTAALLAEIEKQIGLPVTRAVSTHFHDDRV 120
Query 121 GGVDVLRAAGVATYASPSTRRLAEAEGNEIPTHSLEGLSSSGDAVRFGPVELFYPGAAHS 180
GGVDVLRAAGVATYASPSTRRLAE_EGNEIPTHSLEGLSSSGDAVRFGPVELFYPGAAHS
Sbjct 121 GGVDVLRAAGVATYASPSTRRLAEVEGNEIPTHSLEGLSSSGDAVRFGPVELFYPGAAHS 180
Query 181 TDNLVVYVPSANVLYGGCAVHELSSTSAGNVADADLAEWPTSVERIQKHYPEAEVVIPGH 240
TDNLVVYVPSA+VLYGGCA++ELS_TSAGNVADADLAEWPTS+ERIQ+HYPEA+_ VI H
Sbjct 181 TDNLVVYVPSASVLYGGCAIYELSRTSAGNVADADLAEWPTSIERIQQHYPEAQFVIDLH 240
And the amino acid sequence alignment for VIM-2 (NCBI nucleotide accession nr ACH43053) with the study isolate’s VIM-2.
Query: ACH43053.1 Vim-2 [Pseudomonas aeruginosa] Query ID: lcl|Query_6157 Length: 266
>Study_isolate
Sequence ID: Query_6159 Length: 342
Score:478 bits(1231), Expect:7e-177,
Method:Compositional matrix adjust.,
Identities:238/240(99%), Positives:238/240(99%), Gaps:0/240(0%)
Query 1 MFKLLSKLLVYLTASIMAIASPLAFSVDSSGEYPTVSEIPVGEVRLYQIADGVWSHIATQ 60
MFKLLSKLLVYLTASIMAIASPLAFSVDSSGEYPTVSEIPVGEVRLYQIADGVWSHIATQ
Sbjct 1 MFKLLSKLLVYLTASIMAIASPLAFSVDSSGEYPTVSEIPVGEVRLYQIADGVWSHIATQ 60
Query 61 SFDGAVYPSNGLIVRDGDELLLIDTAWGAKNTAALLAEIEKQIGLPVTRAVSTHFHDDRV 120
SFDGAVYPSNGLIVRDGDELLLIDTAWGAKNTAALLAEIEKQIGLPVTRAVSTHFHDDRV
Sbjct 61 SFDGAVYPSNGLIVRDGDELLLIDTAWGAKNTAALLAEIEKQIGLPVTRAVSTHFHDDRV 120
Query 121 GGVDVLRAAGVATYASPSTRRLAEVEGNEIPTHSLEGLSSSGDAVRFGPVELFYPGAAHS 180
GGVDVLRAAGVATYASPSTRRLAEVEGNEIPTHSLEGLSSSGDAVRFGPVELFYPGAAHS
Sbjct 121 GGVDVLRAAGVATYASPSTRRLAEVEGNEIPTHSLEGLSSSGDAVRFGPVELFYPGAAHS 180
Query 181 TDNLVVYVPSASVLYGGCAIYELSRTSAGNVADADLAEWPTSIERIQQHYPEAQFVIPGH 240
TDNLVVYVPSASVLYGGCAIYELSRTSAGNVADADLAEWPTSIERIQQHYPEAQFVI H
Sbjct 181 TDNLVVYVPSASVLYGGCAIYELSRTSAGNVADADLAEWPTSIERIQQHYPEAQFVIDLH 240
Reviewer’s comment 11: The outbreak of this study occurred in the period from February 2006 to June 2007 which is more than 15 years ago. I need the author to give the rationale for becoming late in the publication of these findings. And this has to be reflected in the manuscript title for better clarification for readers as it is an old outbreak in Sweden that is not currently identified.
Authors’ response: When we first submitted our manuscript many years ago, the general opinion was that the environment did not play a major role in the spread of P. aeruginosa, and a limited number of published works indicated that sinks could act as a reservoir. Our suggestion that sinks had played a major role in the Blekinge outbreak was therefore not accepted. During the following five years, one of the authors experienced two more sink-mediated outbreaks, and it was noted that the causing multi-drug resistant Enterobacterales could survive for years in sinks. To remove the risk, a self-disinfecting sink prototype was developed, tested in a clinical microbiology laboratory, adjusted, CE-marked, tested again in an ICU, readjusted, and finally studied in a burn center during 2019-2020. The pandemic swallowed thereafter all the resources, but in February 2023 we had an article ready to be published (see below). It took 15 years, but we have now shown prospectively that bacteria can be transmitted from a sink to a patient and that the bacterial load in sinks can be decreased by using self-disinfecting sinks. The awareness of sinks as reservoirs for Gram-negative bacteria is better today than it was 15 years ago, but it is still on a low level. To explain to the reader the hygienic problem with sinks and offer a solution to it, we submitted the old outbreak manuscript at the same time as the new one about self-disinfecting sinks. We also asked the Editor if he/she could consider to publish them together. Finally, the title of this manuscript has been changed to clarify the age of the outbreak.
Source Control of Gram-negative Bacteria by using Self-Disinfecting Sinks in a Swedish Burn Centre
Maria Gideskog1, Tina Falkeborn2, Jenny Welander2, Åsa Melhus3
1Department of Communicable Disease and Infection Control, and Department of Biomedical and Clinical Sciences, Linköping University, Linköping
2Department of Clinical Microbiology, and Department of Biomedical and Clinical Sciences, Linköping University, Linköping
3Department of Medical Sciences, Section of Clinical Microbiology, Uppsala University, Uppsala, Sweden
Corresponding author: Maria Gideskog, Department of Communicable Disease and Infection Control, and Department of Biomedical and Clinical Sciences, Linköping University, Gasverksgränd 2, SE-581 85 Linköping, Sweden. Telephone: +46 (0)101036851. E-mail: maria.gideskog@regionostergotland.se.
Abstract
Several retrospective studies have identified hospital sinks as a reservoir of Gram-negative bacteria. The aim of this study was to investigate prospectively the bacterial transmission from sinks to patients and if self-disinfecting sinks could reduce this risk. Samples were collected weekly from sinks (self-disinfecting, treated with boiling water, not treated) and patients in the Burn Centre at Linköping University Hospital, Sweden. The antibiotic susceptibility of Gram-negative isolates was tested, and eight randomly chosen patient isolates and their connected sink isolates were subjected to whole genome sequencing (WGS). Of 489 sink samples, 232 (47%) showed growth. The most frequent findings were Stenotrophomonas maltophilia (n=130), Pseudomonas aeruginosa (n=128) and Acinetobacter spp. (n=55). Bacterial growth was observed in 20% of the samplings from the self-disinfecting sinks and in 57% from the sinks treated with boiling water (P=0.0029). WGS recognized one transmission of Escherichia coli sampled from an untreated sink to a patient admitted to the same room. In conclusion, the results showed that sinks can serve as a reservoir of Gram-negative bacteria and that self-disinfecting sinks can reduce the transmission risk. Installing self-disinfecting sinks in intensive care units is an important measure in preventing nosocomial infection among critically ill patients.
Keywords: sink; water-trap; bacterial transmission; self-disinfecting sink; infection control; Pseudomonas aeruginosa; Stenotrophomonas maltophilia; Acinetobacter
Introduction
Healthcare-related infections caused by multidrug-resistant Gram-negative bacteria are medically challenging. Few treatment options are usually available due to the wide and complex range of resistance mechanisms these bacteria carry [1, 2], and, as a consequence, they are associated with an increased financial burden, prolonged hospital stays and an increased mortality [3, 4].
In intensive care units (ICUs), the clinical impact of opportunistic Gram-negative bacteria with multidrug-resistance, such as Acinetobacter baumanii, Enterobacter cloacae, Klebsiella pneumonia, Pseudomonas aeruginosa and Stenotrophomonas maltophilia, is increasing. Medical conditions associated with these bacteria range from colonization of the respiratory and urinary tract to deep and disseminated infections [5-7].
In this context, burn patients represent an especially difficult cohort. A loss of a functioning skin barrier in the form of a third-degree burn, often combined with an inhalation injury and endotracheal intubation, entails a dysfunctional immune system and a high vulnerability to colonization of Gram-negative bacteria. Infections are frequent, and they can lead to everything from melting skin grafts to septic shock and death [8]. The more severe or larger the burn injury is, the more likely is it that an infection will ensue. To prevent serious complications, it is essential to have a proactive approach and treat the infection as early and efficient as possible. Cultures are therefore often regularly performed and repeated courses of antibiotics prescribed. The high selective pressure favours multidrug-resistance, and common bacterial findings are Acinetobacter spp., K. pneumoniae and P. aeruginosa [9-11].
With few exceptions, Gram-negative bacteria are sensitive to desiccation. They are therefore typically found in moist environments, e.g. sinks and their drainage systems. The water traps of sinks constitute a relatively protected environment, which favours the growth of bacteria and production of biofilms [12-14]. Once biofilms have been established, disinfectants cannot fully eradicate them [15]. Through splash water and aerosols, bacteria can be mobilized and transmitted from the sinks to patients. Sinks have been identified as a potential source of infections and outbreaks in ICUs in several reports, but their clinical importance has to some extent been questioned due to the lack of prospective studies available [16].
A burn centre is a complex and stressful care environment [17]. Operations are usually performed in the patient room to avoid moving the patient, and the patient may stay for several months. Thus, the sinks located in the patient rooms are frequently used for other purposes than hand washing. Gram-negative bacteria therefore tend to accumulate in the sinks and their drainage systems. To explore the extent of sink contamination, samples from water traps in sinks at the Burn Centre at Linköping University Hospital, Sweden, were cultured the summer 2018. Growth of clinically relevant Gram-negative bacteria was recorded in all sinks placed in patient rooms and the associated bathrooms. Furthermore, several of the identified species were also observed in blood and wound samples from admitted patients.
The aim of this study was to investigate if it would be possible to reduce the load of Gram-negative bacteria in sinks, and thereby also indirectly the risk of nosocomial infections in a burn centre, by installing newly developed self-disinfecting sinks. The design of the study made it also possible to explore prospectively the transmission of Gram-negative bacteria from sinks to patients.
Materials and Methods
Settings
The Burn Centre at Linköping University Hospital, Östergötland County, Sweden, is one of two units for national highly specialised care of severely burned patients in Sweden. Approximately 100 patients are admitted each year. The catch area is nationwide, but the majority of the patients are referred from the south of Sweden. The unit offers a total of seven single-bed rooms, of which four (rooms 1-4) are equipped for intensive care with a high level of medical monitoring and access to respiratory care. There are two sinks per room, one located in the patient room and the other in the bathroom. The sinks are used for hand washing, cleaning of various medical devices and in the direct patient care.
Since the study material only comprised bacterial isolates and no changes were made in well-established clinical routines, no ethical approval was sought.
Self-disinfecting sinks
The self-disinfecting stainless steel sink (Dissinkfect®, Micropharmics AB and Tunerlux AB, Uppsala, Sweden) used in this study has a built-in heating supply, which heat the wash bowl to 75°C and the water trap to 100°C. It tolerates quick temperature changes and commonly used cleansing or disinfecting agents. By pressing a button placed on the side of the sink for four seconds, the disinfection-process starts and a green LED-indicator shines during the whole 15 min process. It can be stopped at any time, and its length and temperatures can be adjusted according to the requirements. During the study period, self-disinfection was initiated once per each work shift, i.e. three times per 24 h.
Two self-disinfecting sinks, one located in the patient room and one in the bathroom, were installed in room 1. This was the intensive care room most frequently occupied by patients prior to the study. Another intensive care room (room 4) was selected as a comparator, and the sinks in this room were treated with boiling water (3 L each) once a week during the entire study period. The remaining sinks at the centre acted as controls and were not disinfected at any time. All sinks were cleaned daily.
Environmental cultures
To explore the growth of different bacteria in the water trap of sinks over time, environmental samples were collected with ESwabs (Copan Diagnostics Inc. Murrieta, CA, USA) from all 14 patient-associated sinks in the Burn Centre. The sampling started directly after the installation of the self-disinfecting sinks in September 2019 and continued on a weekly basis until April 2020, a total of 35 weeks. Records were kept concerning patient occupancy of each room upon sampling.
The samples were sent to the Department of Clinical Microbiology, Linköping University Hospital, and inoculated onto three different types of media: blood agar, hematin agar and chromogenic urinary tract infection (UTI) agar (Thermo Fisher Scientific, Waltham, MA, USA). Discs (Thermo Fisher Scientific, Waltham, MA, USA) with imipenem (10 µg), trimethoprim-sulfamethoxazole (1.25-23.75 µg), and linezolid (10 µg) was placed on the plates, respectively. The plates were incubated at 35°C for approximately 48 h. Bacteria were identified to the species level with a MALDI Biotyper 3.0 (Bruker Corporation, Karlsruhe, Germany).
Patients
All patients admitted to the Burn Centre during the study period were cultured once a week and upon any clinical sign of infection, according to the routines of the unit. ESwabs (Copan Diagnostics Inc.) were used when sampling from burn wounds. The samples were inoculated onto four different types of media: hematin agar, chromogenic UTI agar, streptococcus agar and chromogenic Staphylococcus aureus (CSA) agar (Thermo Fisher Scientific, Waltham, MA, USA) and incubated at 35°C for approximately 48 h. All Gram-negative isolates were frozen at -70℃ to allow for future genetic analyses. Gram-positive isolates were only frozen if it was indicated in the quality manual of the laboratory. During the patient’s stay, it was recorded in which room the patient was placed. Relocation of patients was avoided, unless a patient was moved from a room equipped for intensive care to a regular room as a result of treatment progress.
Antibiotic susceptibility testing
The antibiotic susceptibility was tested with the disc diffusion method according to the recommendations of EUCAST (www.eucast.org). For environmental Gram-negative bacteria it included cefotaxim, ceftazidim, cefepim, piperacillin-tazobactam, imipenem, meropenem, nalidixic acid, ciprofloxacin, tobramycin and trimethoprim-sulfamethoxazole. For P. aeruginosa the susceptibility testing was limited to ceftazidim, imipenem and meropenem, and for S. maltophilia to trimethoprim-sulfamethoxazole.
Patient isolates were tested when judged clinically relevant and against antibiotics recommended for each species. S. aureus resistant to cefoxitin were further analysed with polymerase chain reaction (PCR) to determine carriage of the nuc gene and mecA gene. All methicillin-resistant S. aureus (MRSA) isolates were subjects for whole genome sequencing (WGS).
An isolate was considered multidrug-resistant if it was resistant to at least three classes of antibiotics, P. aeruginosa and S. maltophilia exempted. Enterobacterales with cefotaxim/ceftazidim/cefepim/meropenem susceptibility patterns raising concerns about extended spectrum beta-lactamase (ESBL)-production were further evaluated fenotypically and/or genotypically, in accordance with local guidelines.
WGS
Eight randomly chosen isolates from the same number of patients, and isolates from water-traps of sinks that belonged to the same species and were connected in space and time to each patient, were subjected to WGS. Furthermore, there were rooms in which a patient was colonized by the same species as the former patient, but there was no growth of this species in the sinks. Five of these patients’ isolates were randomly selected for sequencing, together with five isolates from former patients. The patients had stayed in rooms 1, 2, 4, 5, and 6.
DNA was prepared from 1µL from a single colony of each isolate, using EZ1 DNA Tissue Kit (Qiagen, Germantown, MD, USA), with an included pre-heating step at 95°C and shaking at 350 rpm. Twenty ng of DNA was used for library preparation, using QIAseq FX DNA Library Kit (Qiagen, Germantown, MD, USA) with 8 min of fragmentation time. DNA libraries were sequenced on the MiSeq platform (Illumina, San Diego, CA, USA) with 2 x 300 bp paired-end reads.
Data analysis was performed in CLC Genomics Workbench v. 10.1.1 with the Microbial Genomics Module v. 2.5.1 (Qiagen, Germantown, MD, USA). Multilocus sequence typing (MLST) analysis was performed using the PubMLST (pubmlst.org) scheme for each randomly chosen bacterial species. Read mapping and variant calling were performed against the different reference genomes with NCBI accession numbers NC_008253 (Escherichia coli), NC_018405 (E. cloacae), NC_017548 (P. aeruginosa) and NC_010943 (S. maltophilia), with the following thresholds to call a variant: depth of coverage ≥ 20x, frequency ≥ 90% and Phred score ≥ 20. A quality filter was then applied that retained variants with a sequencing depth of ≥ 20x in all samples and a distance ≥ 10 bp to the next variant, and the resulting variants were used to create single nucleotide polymorphism (SNP) trees and calculate genetic distances between samples. Previous studies suggest that isolates of E. coli and P. aeruginosa with a distance of ≤10 SNPs and ≤37 SNPs, respectively, are likely to belong to the same clone [18]. So far, no studies have suggested SNP thresholds for E. cloacae or S. maltophilia, but an SNP distance of <21 has been considered to support that two bacterial isolates in general have arisen from the same source [19].
Statistical analysis
Fischer’s exact test was used when comparing the culture results from the three groups of sinks (self-disinfecting, treated with boiling water, not treated). A P-value of ≤0.05 was considered statistically significant.
Results
Environmental samples
A total of 489 samples were collected from the water-traps of the sinks during the study period. Of these, 232 samples (47%) showed growth of one or more bacterial species. The three most frequent Gram-negative bacteria were S. maltophilia (n=130), P. aeruginosa (n=128) and Acinetobacter spp (n=55). For more details, see Figure 1. The growth of Gram-positive bacteria consisted mostly of skin flora: coagulase-negative staphylococci (n=24), S. aureus (n=1) and Enterococcus faecalis (n=6).
Bacterial growth in one or both of the self-disinfecting sinks located in room 1 was observed at seven (20%) different sampling occasions. The bacterial load in these thinks were significantly lower than in those treated with boiling water once a week (P=0.0029) and those that were not treated at all (P=< 0.00001). The total number of Gram-negative isolates was eleven and consisted of Acinetobacter spp. (n=5), S. maltophilia (n=4) and P. aeruginosa (n=2).
In the sinks treated with boiling water in room 4, 57 Gram-negative bacterial isolates belonging to seven bacterial genera were collected at 20 (57%) different sampling occasions. The sinks located in the remaining rooms (no disinfection treatment) showed the broadest range of bacterial species and an even higher proportion of bacterial growth (see Figure 1).
Fig. 1. The number of bacterial isolates and type of bacteria are shown per room. Each room has two sinks. The percentage of bacterial growth in correlation with the number of sampling occasions are shown above each room.
The distribution of bacteria in the water traps of the sinks in the patient rooms and the bathrooms varied. In room 1, the majority (91%) of the bacteria were sampled from the bathroom. The corresponding figures for rooms 2-7 were 46%, 39%, 42%, 51%, 54%, and 70%, respectively.
The occupancy of the rooms in the burn centre differed. The room with the highest level of occupancy was room 2. It was occupied by four patients during 29 of the study weeks (83%). In contrast, room 7 was only occupied during two weeks (6%) and by two patients. This was the lowest level of occupancy. The remaining rooms were occupied as follows: room 1 by four patients during 23 weeks (66%), room 3 by seven patients during 18 weeks (51%), room 4 by five patients during 22 weeks (63%), room 5 by eight patients during 17 weeks (49%), and room 6 by six patients during 25 weeks (71%). The bacterial growth in the sinks, in correlation with patient occupancy per room, is shown in Figure 2. In all rooms but room 1, an increased accumulation of bacteria was observed when a patient was admitted to the room.
Fig. 2. The percentage of bacterial growth in correlation with patient occupancy are shown for each room.
The antibiotic susceptibility testing revealed a multidrug-resistant E. coli strain sampled from sinks in room 4. It was ESBL-producing, resistant to cefotaxim, ceftazidim, cefepim, piperacillin-tazobactam, nalidixic acid, ciprofloxacin, tobramycin and trimethoprim-sulfamethoxazole, and had been brought into the unit by the patient staying in the room. It was detected in the sinks for a longer time period. After the patient was discharged, the two sinks were treated with boiling water, and no new patient was admitted until the cultures were negative. P. aeruginosa isolates with resistance to ceftazidim, imipenem and meropenem were observed on different sampling occasions from sinks in rooms 1, 2 and 4. The remaining isolates showed no deviant resistance patterns.
Patient samples
A total of 36 patients were admitted to the Burn Centre during the study period. The duration of the stay varied depending on the severity of the burn injuries, e.g. room 2 was occupied by the same patient during 20 weeks before relocation, whereas another patient stayed for less than one week in room 5.
Culture samples collected from the patients showed the following growth of Gram-negatives: P. aeruginosa (n=31), E. cloacae (n=28), E. coli (n=11), Klebsiella spp. (n=7), Proteus spp. (n=6), S. maltophilia (n=6), Acinetobacter spp. (n=3), Serratia marcescens (n=2), Citrobacter freundii (n=2), Morganella morganii (n=1), Enterobacter amnigenus (n=1) and Moraxella catarrhalis (n=1). The growth of Gram-positive bacteria consisted of S. aureus (n=274), coagulase-negative staphylococci (n=88), Enterococcus spp (n=89), Streptococcus spp (n=37) and Bacillus spp (n=13).
Multidrug-resistant bacteria isolated from patients included seven samples of MRSA isolated from two patients admitted to room 6 on different occasions (unrelated strains and never found in any sink), and the multidrug-resistant E. coli found in the sinks in room 4. It was isolated from the patient at admittance. A P. aeruginosa strain with resistance to ceftazidim, ciprofloxacin, imipenem, meropenem and piperacillin/tazobactam was isolated at several different sampling occasions from a patient admitted to room 2. This patient was in addition colonized by an E. cloacae strain resistant to ceftazidim, cefotaxim and piperacillin/tazobactam.
WGS results
A total of 24 isolates were subjected to WGS: E. cloacae complex (n=8), P. aeruginosa (n=8), E. coli (n=4) and S. maltophilia (n=4). Two of the P. aeruginosa genomes were used twice, i.e. they were not only included when comparing sink-patient genomes but also when comparing patient-patient genomes when there was no growth of the bacterium in the sinks.
The samples obtained an average sequencing depth of 64 x. One cluster was recognized with MLST and whole genome-wide phylogenetic analysis. The cluster contained two isolates of E. coli, one sampled from a patient placed in room 6 and the other was an environmental sample collected from one of the sinks in the same room one week earlier. The isolates had a difference of a single SNP and were identified as sequence type (ST) 625. The remaining isolates belonged all to unique clones.
Isolates identified with MALDI-TOF mass spectrometry as E. cloacae complex constituted a special problem. In half of the cases, the genomes that were compared did not belong to the same species. Species within the complex identified with WGS included Enterobacter roggenkampii, Enterobacter hormaechei and Enterobacter ludwigii.
The E. cloacae complex is known for its ability to harbour the plasmid-mediated sil operon, a gene cluster encoding efflux pumps, a silver-binding protein and regulatory genes that confers resistance to silver [20]. Silver products are often used in burn centres and could therefore select for this bacterial complex, which was a relatively frequent finding in both sink and patient samples. The genomes of isolates belonging to the E. cloacae complex were therefore screened for the sil operon. Six out of eight isolates (75%) carried the full operon.
Discussion
There has been a clear increase of sink-associated outbreaks caused by Gram-negative bacteria in late years [21-26]. In the present study, it was investigated if stainless steel sinks, in which both the bowl and the water-trap were self-disinfected three times per 24 h, could reduce the bacterial load and thereby the risk of transmission. Furthermore, two conventional sinks were treated weekly with boiling water as an easy and cheaper alternative. The results showed that both alternatives reduced the bacterial load of the sinks compared with no disinfection at all, but the self-disinfecting sinks were significantly more efficient. This is in accord with other studies in which self-disinfecting sink drains have been used [27, 28].
The self-disinfecting sinks in room 1 had the overall lowest frequency of bacterial growth and the fewest number of species isolated during the entire study period. In contrast to all the other rooms, there was no correlation between patient occupancy and the bacterial growth in the sinks; the bacterial load remained low or was zero despite a suboptimal use of the sinks during patient care. Although the routine to initiate a self-disinfecting cycle every 8 h did not eliminate all bacterial growth, it showed that it could be radically decreased. It is quite possible that the bacterial growth would have been further reduced if the self-disinfecting cycle had been started every time the sink was contaminated. The health care personnel at the centre found, however, this instruction too complicated and time-consuming, why it was changed to every 8 h.
Treatment with boiling water was a simple and functioning alternative that kept the bacterial load at a relatively low level in the sinks located in room 4. As shown in a study from 2021, the initial concentration of bacteria in the drain is back within approximately a week [29]. Thus, this alternative needs to be carried out at least once weekly and continuously to avoid re-occurrence of growth. In addition, the procedure involves an extra workload for the personnel, and there is always a risk of contracting burn injuries while handling the boiling water. It was, however, chosen over chlorine, the traditional disinfectant for hospital sinks, since it has been shown to be 100 to 1000 times more effective in reducing pathogens, it does not smell, it is environmentally friendly and fairly inexpensive [29]. Replacement of contaminated sinks has been shown to reduce the infection rates in ICUs [30, 31], but bacteria may not only reside in the water-trap. They can also be found further down in the drainage system. As a result, bacteria can reappear despite a complete change of sinks [24]. Self-disinfecting sinks are therefore a better and the most long-term and solution to the problem.
The whole genome-wide phylogenetic analysis identified one cluster among the 24 patient and environmental samples that were randomly chosen. The cluster consisted of two E. coli isolates belonging to ST625, a clone associated with extra-intestinal infections [32]. The sink isolate was collected one week earlier than the patient isolate, indicating that the sink was the likely source of the bacterium that colonized the patient’s burn wounds. This is, to our knowledge, the first time this type of event has been observed prospectively in a clinical setting. The exact route for the transmission is, however, not clear. Few studies deal with the exact mechanism of transmission from a sink to a patient. In a recent study, mobilization of bacteria from biofilms in water-traps of sinks to the surrounding environment was demonstrated by using green fluorescent-expressing E. coli [12]. This is a possible transmission route for the E. coli in room 6.
Additional transmissions may also have occurred in this and other rooms, but the low number of isolates investigated and the fact that only a single colony was used when preparing the DNA limited the chances of detecting them. Interestingly, in the five cases where a patient was colonized by the same Gram-negative species as the former patient, and the sinks lacked growth of the species of interest, no transmission was observed. However, the number of colonies/isolates investigated can once again have been too low.
There were few multidrug-resistant isolates in the present study, but resistance does not always come in the form of antibiotic resistance. The isolation frequency of E. cloacae complex was relatively high among the Gram-negatives. Only one isolate was resistant to more broad-spectrum beta-lactams, whereas the carriage rate of the sil operon was quite high, at 75%. In an earlier study [20], 48% of invasive E. cloacae isolates harboured sil genes. These findings suggest that the use of silver products rather than antibiotics could have selected for this complex, but if the genes were expressed or not was never tested.
Although the main focus in this study was on Gram-negative bacteria, it was striking how few Gram-positive bacteria were isolated from the sinks compared with from the patients. For instance, only a single S. aureus isolate was recorded from the sinks. The corresponding figure from patients was 274, indicating that water traps offer mainly an environment that promotes growth of Gram-negative bacteria, and of S. maltophilia and P. aeruginosa in particular. However, even if S. aureus did not thrive in the water traps, it may survive, together with other Gram-positive bacteria and Acinetobacter spp., in the wash bowl. To reduce the risk of dissemination from this part of the sink, the wash bowl was also decontaminated during the disinfection process.
In conclusion, the results showed prospectively that sinks can serve as a reservoir of Gram-negative bacteria, and that self-disinfecting sinks can reduce the bacterial load in the sinks and thereby also the risk of bacterial transmission. Installing self-disinfecting sinks in ICUs is therefore an important measure in preventing nosocomial infection among critically ill and vulnerable patients. A less expensive but not as efficient solution can be to disinfect sinks with boiling water once weekly.
Funding
This study was financially supported by the National Association Heart-Lung and ALF-funds.
Acknowledgements
We thank the staff at the Burn Centre at Linköping University Hospital, Sweden, for their excellent help during the study period.
Author Contributions
Conceptualization, visualization, and data interpretation: M.G., Å.M.; Methodology and Analysis: M.G., T.F., J.W., Å.M.; Funding and Resources: Å.M.; Writing – original and revised draft preparation: M.G., J.W., Å.M. All authors have read and agreed to the published version of the manuscript.
Conflicts of Interest
The self-disinfecting sink was invented by Micropharmics AB (owned by Å.M.) and developed together with Tunerlux AB. The funder had no role in study design, data collection and interpretation, or the decision to submit the work for publication.
References
- Coppola, N.; Maraolo, A.E.; Onorato, L.; Scotto, R.; Calo, F.; Atripaldi, L.; Borrelli, A.; Corcione, A.; De Cristofaro, M.G.; Durante-Mangoni, E.; Filippelli, A.; Franci, G.; Galdo, M.; Guglielmi, G.; Pagliano, P.; Perrella, A.; Piazza, O.; Picardi, M.; Punzi, R.; Trama, U.; Gentile, I. Epidemiology, Mechanisms of Resistance and Treatment Algorithm for Infections Due to Carbapenem-Resistant Gram-Negative Bacteria: An Expert Panel Opinion. Antibiotics (Basel). 2022, 11. doi: 10.3390/antibiotics11091263.
- Wagner, T.M.; Howden, B.P.; Sundsfjord, A.; Hegstad, K. Transiently silent acquired antimicrobial resistance: an emerging challenge in susceptibility testing. J Antimicrob Chemother. 2023, 78, 586-98. doi: 10.1093/jac/dkad024.
- Serra-Burriel, M.; Keys, M.; Campillo-Artero, C.; Agodi, A.; Barchitta, M.; Gikas, A.; Palos, C.; Lopez-Casasnovas, G. Impact of multi-drug resistant bacteria on economic and clinical outcomes of healthcare-associated infections in adults: Systematic review and meta-analysis. PLoS One. 2020, 15, e0227139. doi: 10.1371/journal.pone.0227139.
- Antimicrobial Resistance C. Global burden of bacterial antimicrobial resistance in 2019: a systematic analysis. Lancet. 2022, 399, 629-55. doi: 10.1016/S0140-6736(21)02724-0.
- MacVane, S.H. Antimicrobial Resistance in the Intensive Care Unit: A Focus on Gram-Negative Bacterial Infections. J Intensive Care Med. 2017, 32, 25-37. doi: 10.1177/0885066615619895.
- Sader, H.S.; Mendes, R.E.; Streit, J.M.; Carvalhaes, C.G.; Castanheira, M. Antimicrobial susceptibility of Gram-negative bacteria from intensive care unit and non-intensive care unit patients from United States hospitals (2018-2020). Diagn Microbiol Infect Dis. 2022, 102, 115557. doi: 10.1016/j.diagmicrobio.2021.115557.
- Decker, B.K.; Palmore, T.N. The role of water in healthcare-associated infections. Curr Opin Infect Dis. 2013, 26, 345-51. doi: 10.1097/QCO.0b013e3283630adf.
- Ladhani, H.A.; Yowler, C.J.; Claridge, J.A. Burn Wound Colonization, Infection, and Sepsis. Surg Infect (Larchmt). 2021, 22, 44-8. doi: 10.1089/sur.2020.346.
- Yin, Z.; Beiwen, W.; Zhenzhu, M.; Erzhen, C.; Qin, Z.; Yi, D. Characteristics of bloodstream infection and initial antibiotic use in critically ill burn patients and their impact on patient prognosis. Sci Rep. 2022, 12, 20105. doi: 10.1038/s41598-022-24492-z.
- Haider, M.H.; McHugh, T.D.; Roulston, K.; Arruda, L.B.; Sadouki, Z.; Riaz, S. Detection of carbapenemases bla(OXA48)-bla(KPC)-bla(NDM)-bla(VIM) and extended-spectrum-beta-lactamase bla(OXA1)-bla(SHV)-bla(TEM) genes in Gram-negative bacterial isolates from ICU burns patients. Ann Clin Microbiol Antimicrob. 2022, 21, 18. doi: 10.1186/s12941-022-00510-w.
- Lima, W.G.; Silva Alves, G.C.; Sanches, C.; Antunes Fernandes, S.O.; de Paiva, M.C. Carbapenem-resistant Acinetobacter baumannii in patients with burn injury: A systematic review and meta-analysis. Burns. 2019, 45, 1495-508. doi: 10.1016/j.burns.2019.07.006.
- Kotay, S.; Chai, W.; Guilford, W.; Barry, K.; Mathers, A.J. Spread from the Sink to the Patient: In Situ Study Using Green Fluorescent Protein (GFP)-Expressing Escherichia coli To Model Bacterial Dispersion from Hand-Washing Sink-Trap Reservoirs. Appl Environ Microbiol. 2017, 83. doi: 10.1128/AEM.03327-16.
- Valentin, A.S.; Santos, S.D.; Goube, F.; Gimenes, R.; Decalonne, M.; Mereghetti, L.; Daniau, C.; van der Mee-Marquet, N. A prospective multicentre surveillance study to investigate the risk associated with contaminated sinks in the intensive care unit. Clin Microbiol Infect. 2021, 27, 1347 e9- e14. doi: 10.1016/j.cmi.2021.02.018.
- Kearney, A.; Boyle, M.A.; Curley, G.F.; Humphreys, H. Preventing infections caused by carbapenemase-producing bacteria in the intensive care unit - Think about the sink. J Crit Care. 2021, 66, 52-9. doi: 10.1016/j.jcrc.2021.07.023.
- Loveday, H.P.; Wilson, J.A.; Kerr, K.; Pitchers, R.; Walker, J.T.; Browne, J. Association between healthcare water systems and Pseudomonas aeruginosa infections: a rapid systematic review. J Hosp Infect. 2014, 86, 7-15. doi: 10.1016/j.jhin.2013.09.010.
- Volling, C.; Ahangari, N.; Bartoszko, J.J.; Coleman, B.L.; Garcia-Jeldes, F.; Jamal, A.J.; Johnstone, J.; Kandel, C.; Kohler, P.; Maltezou, H.; Maze Dit Mieusement, L.; McKenzie, N.; Mertz, D.; Monod, A.; Saeed, S.; Shea, B.; Stuart, R.; Thomas, S.; Uleryk, E.; McGeer, A. Are Sink Drainage Systems a Reservoir for Hospital-Acquired Gammaproteobacteria Colonization and Infection? A Systematic Review. Open Forum Infect Dis. 2021, 8, ofaa590. doi: 10.1093/ofid/ofaa590.
- Bayuo, J.; Agbenorku, P. Coping strategies among nurses in the Burn Intensive Care Unit: A qualitative study. Burns Open. 2018, 2, 47-52. doi: 10.1016/J.BURNSO.2017.10.004.
- Schurch, A.C.; Arredondo-Alonso, S.; Willems, R.J.L.; Goering, R.V. Whole genome sequencing options for bacterial strain typing and epidemiologic analysis based on single nucleotide polymorphism versus gene-by-gene-based approaches. Clin Microbiol Infect. 2018, 24, 350-4. doi: 10.1016/j.cmi.2017.12.016.
- Pightling, A.W.; Pettengill, J.B.; Luo, Y.; Baugher, J.D.; Rand, H.; Strain, E. Interpreting Whole-Genome Sequence Analyses of Foodborne Bacteria for Regulatory Applications and Outbreak Investigations. Front Microbiol. 2018, 9, 1482. doi: 10.3389/fmicb.2018.01482.
- Sutterlin, S.; Dahlo, M.; Tellgren-Roth, C.; Schaal, W.; Melhus, A. High frequency of silver resistance genes in invasive isolates of Enterobacter and Klebsiella species. J Hosp Infect. 2017, 96, 256-61. doi: 10.1016/j.jhin.2017.04.017.
- Gideskog, M.; Welander, J.; Melhus, A. Cluster of S. maltophilia among patients with respiratory tract infections at an intensive care unit. Infect Prev Pract. 2020, 2, 100097. doi: 10.1016/j.infpip.2020.100097.
- Jung, J.; Choi, H.S.; Lee, J.Y.; Ryu, S.H.; Kim, S.K.; Hong, M.J.; Kwak, S.H.; Kim, H.J.; Lee, M.S.; Sung, H.; Kim, M.N.; Kim, S.H. Outbreak of carbapenemase-producing Enterobacteriaceae associated with a contaminated water dispenser and sink drains in the cardiology units of a Korean hospital. J Hosp Infect. 2020,104, 476-83. doi: 10.1016/j.jhin.2019.11.015.
- Catho, G.; Martischang, R.; Boroli, F.; Chraiti, M.N.; Martin, Y.; Koyluk Tomsuk, Z.; Renzi, G.; Schrenzel, J.; Pugin, J.; Nordmann, P.; Blanc, D.S.; Harbarth, S. Outbreak of Pseudomonas aeruginosa producing VIM carbapenemase in an intensive care unit and its termination by implementation of waterless patient care. Crit Care. 2021, 25, 301. doi: 10.1186/s13054-021-03726-y.
- Stjarne Aspelund, A.; Sjostrom, K.; Olsson Liljequist, B.; Morgelin, M.; Melander, E.; Pahlman, L.I. Acetic acid as a decontamination method for sink drains in a nosocomial outbreak of metallo-beta-lactamase-producing Pseudomonas aeruginosa. J Hosp Infect. 2016, 94, 13-20. doi: 10.1016/j.jhin.2016.05.009.
- Rehou, S.; Rotman, S.; Avaness, M.; Salt, N.; Jeschke, M.G.; Shahrokhi, S. Outbreak of carbapenemase-producing Enterobacteriaceae in a regional burn centre. J Burn Care Res. 2022, 43, 1203-6. doi: 10.1093/jbcr/irac067.
- Salm, F.; Deja, M.; Gastmeier, P.; Kola, A.; Hansen, S.; Behnke, M.; Gruhl, D.; Leistner, R. Prolonged outbreak of clonal MDR Pseudomonas aeruginosa on an intensive care unit: contaminated sinks and contamination of ultra-filtrate bags as possible route of transmission? Antimicrob Resist Infect Control. 2016, 5, 53. doi: 10.1186/s13756-016-0157-9.
- de Jonge, E.; de Boer, M.G.J.; van Essen, E.H.R.; Dogterom-Ballering, H.C.M.; Veldkamp, K.E. Effects of a disinfection device on colonization of sink drains and patients during a prolonged outbreak of multidrug-resistant Pseudomonas aeruginosa in an intensive care unit. J Hosp Infect. 2019, 102, 70-4. doi: 10.1016/j.jhin.2019.01.003.
- Fusch, C.; Pogorzelski, D.; Main, C.; Meyer, C.L.; El Helou, S.; Mertz, D. Self-disinfecting sink drains reduce the Pseudomonas aeruginosa bioburden in a neonatal intensive care unit. Acta Paediatr. 2015, 104, e344-9. doi: 10.1111/apa.13005.
- Diamond, F. Hot water might disinfect sinks better than chlorine. Infection control today. Available online: https://www.infectioncontroltoday.com/view/hot-water-might-disinfect-sinks-better-than-chlorine (accessed on 15 November 2022).
- Hota, S.; Hirji, Z.; Stockton, K.; Lemieux, C.; Dedier, H.; Wolfaardt, G.; Gardam, M. Outbreak of multidrug-resistant Pseudomonas aeruginosa colonization and infection secondary to imperfect intensive care unit room design. Infect Control Hosp Epidemiol. 2009, 30, 25-33. doi: 10.1086/592700.
- Longtin, Y.; Troillet, N.; Touveneau, S.; Boillat, N.; Rimensberger, P.; Dharan, S.; Gervaix, A.; Pittet, D.; Harbarth, S. Pseudomonas aeruginosa outbreak in a pediatric intensive care unit linked to a humanitarian organization residential center. Pediatr Infect Dis J. 2010, 29, 233-7. doi: 10.1097/INF.0b013e3181bc24fb.
- Alonso, C.A.; Gonzalez-Barrio, D.; Ruiz-Fons, F.; Ruiz-Ripa, L.; Torres, C. High frequency of B2 phylogroup among non-clonally related fecal Escherichia coli isolates from wild boars, including the lineage ST131. FEMS Microbiol Ecol. 2017, 93. doi: 10.1093/femsec/fix016.
Author Response File
Reviewer 2 Report
I think that this paper is a well-written paper that traces the origin of VIM-2-producing Pseudomonas aeruginosa in various ways and presents a solution. However, the biggest question is why the study on the outbreak, which occurred in 2006, is only now being published. An explanation is needed for this, and it seems that there should be sufficient discussion about what the research on outbreak means after a considerable time has passed at this point.
In addition, whole genome sequencing was performed on only one strain, but it is doubtful whether this is sufficient, and PFGE results are not presented.
Author Response
Reviewer 2
Reviewer’s comment: I think that this paper is a well-written paper that traces the origin of VIM-2-producing Pseudomonas aeruginosa in various ways and presents a solution. However, the biggest question is why the study on the outbreak, which occurred in 2006, is only now being published. An explanation is needed for this, and it seems that there should be sufficient discussion about what the research on outbreak means after a considerable time has passed at this point.
Authors’ response: When we first submitted our manuscript many years ago, the general opinion was that the environment did not play a major role in the spread of P. aeruginosa, and a limited number of published works indicated that sinks could act as a reservoir. Our suggestion that sinks had played a major role in the Blekinge outbreak was therefore not accepted. During the following five years, one of the authors experienced two more sink-mediated outbreaks, and it was noted that the causing multi-drug resistant Enterobacterales could survive for years in sinks. To remove the risk, a self-disinfecting sink prototype was developed, tested in a clinical microbiology laboratory, adjusted, CE-marked, tested again in an ICU, readjusted, and finally studied in a burn center during 2019-2020. The pandemic swallowed thereafter all the resources, but in February 2023 the article was ready to be published (see below). It took 15 years, but we have now been able to show prospectively that bacteria can be transmitted from a sink to a patient and that the bacterial load in sinks can be decreased by using self-disinfecting sinks. The awareness of sinks as reservoirs for Gram-negative bacteria is better today than it was 15 years ago, but it is still on a low level. To explain to the reader the hygienic problem with sinks and offer a solution to it, we submitted the old outbreak manuscript at the same time as the new one about transmission and self-disinfecting sinks (see below). We also asked the Editor if he/she could consider to publish them together. Apart from being asked more than once to complete the picture of the dissemination of the VIM-2-producing P. aeruginosa strain in the southern part of Sweden, we do think that there can be a value in publishing old outbreaks in a context like this.
Reviewer’s comment: In addition, whole genome sequencing was performed on only one strain, but it is doubtful whether this is sufficient, and PFGE results are not presented.
Authors’ response: When the outbreak took place, WGS was extremely expensive (about 15,000 EUR/isolate) and used only by a few. When the isolate from patient 3 was finally sequenced, the outbreak in Blekinge was long over. The reason why this randomly chosen isolate was sequenced at all, was to clarify its relationship to the outbreak strain in Lund. Since the WGS result would have no impact on any of the outbreaks, the financial support was a minimum. Although WGS is definitely better at discriminating between isolates than PFGE, there is no consensus when two P. aeruginosa isolates can be considered to have the same origin. The suggested SNP difference is currently 37 SNPs or less, which is a relatively big difference. Furthermore, the number of articles supporting this SNP difference is quite low. We are therefore not fully convinced that the old reference method, PFGE, can be totally disqualified in this case. It would have been nice to have all isolates sequenced, but to accomplish this within 10 days, the time given for the revision, is impossible. We have, however, included a photo of the PFGE.
Source Control of Gram-negative Bacteria by using Self-Disinfecting Sinks in a Swedish Burn Centre
Maria Gideskog1, Tina Falkeborn2, Jenny Welander2, Åsa Melhus3
1Department of Communicable Disease and Infection Control, and Department of Biomedical and Clinical Sciences, Linköping University, Linköping
2Department of Clinical Microbiology, and Department of Biomedical and Clinical Sciences, Linköping University, Linköping
3Department of Medical Sciences, Section of Clinical Microbiology, Uppsala University, Uppsala, Sweden
Corresponding author: Maria Gideskog, Department of Communicable Disease and Infection Control, and Department of Biomedical and Clinical Sciences, Linköping University, Gasverksgränd 2, SE-581 85 Linköping, Sweden. Telephone: +46 (0)101036851. E-mail: maria.gideskog@regionostergotland.se.
Abstract
Several retrospective studies have identified hospital sinks as a reservoir of Gram-negative bacteria. The aim of this study was to investigate prospectively the bacterial transmission from sinks to patients and if self-disinfecting sinks could reduce this risk. Samples were collected weekly from sinks (self-disinfecting, treated with boiling water, not treated) and patients in the Burn Centre at Linköping University Hospital, Sweden. The antibiotic susceptibility of Gram-negative isolates was tested, and eight randomly chosen patient isolates and their connected sink isolates were subjected to whole genome sequencing (WGS). Of 489 sink samples, 232 (47%) showed growth. The most frequent findings were Stenotrophomonas maltophilia (n=130), Pseudomonas aeruginosa (n=128) and Acinetobacter spp. (n=55). Bacterial growth was observed in 20% of the samplings from the self-disinfecting sinks and in 57% from the sinks treated with boiling water (P=0.0029). WGS recognized one transmission of Escherichia coli sampled from an untreated sink to a patient admitted to the same room. In conclusion, the results showed that sinks can serve as a reservoir of Gram-negative bacteria and that self-disinfecting sinks can reduce the transmission risk. Installing self-disinfecting sinks in intensive care units is an important measure in preventing nosocomial infection among critically ill patients.
Keywords: sink; water-trap; bacterial transmission; self-disinfecting sink; infection control; Pseudomonas aeruginosa; Stenotrophomonas maltophilia; Acinetobacter
Introduction
Healthcare-related infections caused by multidrug-resistant Gram-negative bacteria are medically challenging. Few treatment options are usually available due to the wide and complex range of resistance mechanisms these bacteria carry [1, 2], and, as a consequence, they are associated with an increased financial burden, prolonged hospital stays and an increased mortality [3, 4].
In intensive care units (ICUs), the clinical impact of opportunistic Gram-negative bacteria with multidrug-resistance, such as Acinetobacter baumanii, Enterobacter cloacae, Klebsiella pneumonia, Pseudomonas aeruginosa and Stenotrophomonas maltophilia, is increasing. Medical conditions associated with these bacteria range from colonization of the respiratory and urinary tract to deep and disseminated infections [5-7].
In this context, burn patients represent an especially difficult cohort. A loss of a functioning skin barrier in the form of a third-degree burn, often combined with an inhalation injury and endotracheal intubation, entails a dysfunctional immune system and a high vulnerability to colonization of Gram-negative bacteria. Infections are frequent, and they can lead to everything from melting skin grafts to septic shock and death [8]. The more severe or larger the burn injury is, the more likely is it that an infection will ensue. To prevent serious complications, it is essential to have a proactive approach and treat the infection as early and efficient as possible. Cultures are therefore often regularly performed and repeated courses of antibiotics prescribed. The high selective pressure favours multidrug-resistance, and common bacterial findings are Acinetobacter spp., K. pneumoniae and P. aeruginosa [9-11].
With few exceptions, Gram-negative bacteria are sensitive to desiccation. They are therefore typically found in moist environments, e.g. sinks and their drainage systems. The water traps of sinks constitute a relatively protected environment, which favours the growth of bacteria and production of biofilms [12-14]. Once biofilms have been established, disinfectants cannot fully eradicate them [15]. Through splash water and aerosols, bacteria can be mobilized and transmitted from the sinks to patients. Sinks have been identified as a potential source of infections and outbreaks in ICUs in several reports, but their clinical importance has to some extent been questioned due to the lack of prospective studies available [16].
A burn centre is a complex and stressful care environment [17]. Operations are usually performed in the patient room to avoid moving the patient, and the patient may stay for several months. Thus, the sinks located in the patient rooms are frequently used for other purposes than hand washing. Gram-negative bacteria therefore tend to accumulate in the sinks and their drainage systems. To explore the extent of sink contamination, samples from water traps in sinks at the Burn Centre at Linköping University Hospital, Sweden, were cultured the summer 2018. Growth of clinically relevant Gram-negative bacteria was recorded in all sinks placed in patient rooms and the associated bathrooms. Furthermore, several of the identified species were also observed in blood and wound samples from admitted patients.
The aim of this study was to investigate if it would be possible to reduce the load of Gram-negative bacteria in sinks, and thereby also indirectly the risk of nosocomial infections in a burn centre, by installing newly developed self-disinfecting sinks. The design of the study made it also possible to explore prospectively the transmission of Gram-negative bacteria from sinks to patients.
Materials and Methods
Settings
The Burn Centre at Linköping University Hospital, Östergötland County, Sweden, is one of two units for national highly specialised care of severely burned patients in Sweden. Approximately 100 patients are admitted each year. The catch area is nationwide, but the majority of the patients are referred from the south of Sweden. The unit offers a total of seven single-bed rooms, of which four (rooms 1-4) are equipped for intensive care with a high level of medical monitoring and access to respiratory care. There are two sinks per room, one located in the patient room and the other in the bathroom. The sinks are used for hand washing, cleaning of various medical devices and in the direct patient care.
Since the study material only comprised bacterial isolates and no changes were made in well-established clinical routines, no ethical approval was sought.
Self-disinfecting sinks
The self-disinfecting stainless steel sink (Dissinkfect®, Micropharmics AB and Tunerlux AB, Uppsala, Sweden) used in this study has a built-in heating supply, which heat the wash bowl to 75°C and the water trap to 100°C. It tolerates quick temperature changes and commonly used cleansing or disinfecting agents. By pressing a button placed on the side of the sink for four seconds, the disinfection-process starts and a green LED-indicator shines during the whole 15 min process. It can be stopped at any time, and its length and temperatures can be adjusted according to the requirements. During the study period, self-disinfection was initiated once per each work shift, i.e. three times per 24 h.
Two self-disinfecting sinks, one located in the patient room and one in the bathroom, were installed in room 1. This was the intensive care room most frequently occupied by patients prior to the study. Another intensive care room (room 4) was selected as a comparator, and the sinks in this room were treated with boiling water (3 L each) once a week during the entire study period. The remaining sinks at the centre acted as controls and were not disinfected at any time. All sinks were cleaned daily.
Environmental cultures
To explore the growth of different bacteria in the water trap of sinks over time, environmental samples were collected with ESwabs (Copan Diagnostics Inc. Murrieta, CA, USA) from all 14 patient-associated sinks in the Burn Centre. The sampling started directly after the installation of the self-disinfecting sinks in September 2019 and continued on a weekly basis until April 2020, a total of 35 weeks. Records were kept concerning patient occupancy of each room upon sampling.
The samples were sent to the Department of Clinical Microbiology, Linköping University Hospital, and inoculated onto three different types of media: blood agar, hematin agar and chromogenic urinary tract infection (UTI) agar (Thermo Fisher Scientific, Waltham, MA, USA). Discs (Thermo Fisher Scientific, Waltham, MA, USA) with imipenem (10 µg), trimethoprim-sulfamethoxazole (1.25-23.75 µg), and linezolid (10 µg) was placed on the plates, respectively. The plates were incubated at 35°C for approximately 48 h. Bacteria were identified to the species level with a MALDI Biotyper 3.0 (Bruker Corporation, Karlsruhe, Germany).
Patients
All patients admitted to the Burn Centre during the study period were cultured once a week and upon any clinical sign of infection, according to the routines of the unit. ESwabs (Copan Diagnostics Inc.) were used when sampling from burn wounds. The samples were inoculated onto four different types of media: hematin agar, chromogenic UTI agar, streptococcus agar and chromogenic Staphylococcus aureus (CSA) agar (Thermo Fisher Scientific, Waltham, MA, USA) and incubated at 35°C for approximately 48 h. All Gram-negative isolates were frozen at -70℃ to allow for future genetic analyses. Gram-positive isolates were only frozen if it was indicated in the quality manual of the laboratory. During the patient’s stay, it was recorded in which room the patient was placed. Relocation of patients was avoided, unless a patient was moved from a room equipped for intensive care to a regular room as a result of treatment progress.
Antibiotic susceptibility testing
The antibiotic susceptibility was tested with the disc diffusion method according to the recommendations of EUCAST (www.eucast.org). For environmental Gram-negative bacteria it included cefotaxim, ceftazidim, cefepim, piperacillin-tazobactam, imipenem, meropenem, nalidixic acid, ciprofloxacin, tobramycin and trimethoprim-sulfamethoxazole. For P. aeruginosa the susceptibility testing was limited to ceftazidim, imipenem and meropenem, and for S. maltophilia to trimethoprim-sulfamethoxazole.
Patient isolates were tested when judged clinically relevant and against antibiotics recommended for each species. S. aureus resistant to cefoxitin were further analysed with polymerase chain reaction (PCR) to determine carriage of the nuc gene and mecA gene. All methicillin-resistant S. aureus (MRSA) isolates were subjects for whole genome sequencing (WGS).
An isolate was considered multidrug-resistant if it was resistant to at least three classes of antibiotics, P. aeruginosa and S. maltophilia exempted. Enterobacterales with cefotaxim/ceftazidim/cefepim/meropenem susceptibility patterns raising concerns about extended spectrum beta-lactamase (ESBL)-production were further evaluated fenotypically and/or genotypically, in accordance with local guidelines.
WGS
Eight randomly chosen isolates from the same number of patients, and isolates from water-traps of sinks that belonged to the same species and were connected in space and time to each patient, were subjected to WGS. Furthermore, there were rooms in which a patient was colonized by the same species as the former patient, but there was no growth of this species in the sinks. Five of these patients’ isolates were randomly selected for sequencing, together with five isolates from former patients. The patients had stayed in rooms 1, 2, 4, 5, and 6.
DNA was prepared from 1µL from a single colony of each isolate, using EZ1 DNA Tissue Kit (Qiagen, Germantown, MD, USA), with an included pre-heating step at 95°C and shaking at 350 rpm. Twenty ng of DNA was used for library preparation, using QIAseq FX DNA Library Kit (Qiagen, Germantown, MD, USA) with 8 min of fragmentation time. DNA libraries were sequenced on the MiSeq platform (Illumina, San Diego, CA, USA) with 2 x 300 bp paired-end reads.
Data analysis was performed in CLC Genomics Workbench v. 10.1.1 with the Microbial Genomics Module v. 2.5.1 (Qiagen, Germantown, MD, USA). Multilocus sequence typing (MLST) analysis was performed using the PubMLST (pubmlst.org) scheme for each randomly chosen bacterial species. Read mapping and variant calling were performed against the different reference genomes with NCBI accession numbers NC_008253 (Escherichia coli), NC_018405 (E. cloacae), NC_017548 (P. aeruginosa) and NC_010943 (S. maltophilia), with the following thresholds to call a variant: depth of coverage ≥ 20x, frequency ≥ 90% and Phred score ≥ 20. A quality filter was then applied that retained variants with a sequencing depth of ≥ 20x in all samples and a distance ≥ 10 bp to the next variant, and the resulting variants were used to create single nucleotide polymorphism (SNP) trees and calculate genetic distances between samples. Previous studies suggest that isolates of E. coli and P. aeruginosa with a distance of ≤10 SNPs and ≤37 SNPs, respectively, are likely to belong to the same clone [18]. So far, no studies have suggested SNP thresholds for E. cloacae or S. maltophilia, but an SNP distance of <21 has been considered to support that two bacterial isolates in general have arisen from the same source [19].
Statistical analysis
Fischer’s exact test was used when comparing the culture results from the three groups of sinks (self-disinfecting, treated with boiling water, not treated). A P-value of ≤0.05 was considered statistically significant.
Results
Environmental samples
A total of 489 samples were collected from the water-traps of the sinks during the study period. Of these, 232 samples (47%) showed growth of one or more bacterial species. The three most frequent Gram-negative bacteria were S. maltophilia (n=130), P. aeruginosa (n=128) and Acinetobacter spp (n=55). For more details, see Figure 1. The growth of Gram-positive bacteria consisted mostly of skin flora: coagulase-negative staphylococci (n=24), S. aureus (n=1) and Enterococcus faecalis (n=6).
Bacterial growth in one or both of the self-disinfecting sinks located in room 1 was observed at seven (20%) different sampling occasions. The bacterial load in these thinks were significantly lower than in those treated with boiling water once a week (P=0.0029) and those that were not treated at all (P=< 0.00001). The total number of Gram-negative isolates was eleven and consisted of Acinetobacter spp. (n=5), S. maltophilia (n=4) and P. aeruginosa (n=2).
In the sinks treated with boiling water in room 4, 57 Gram-negative bacterial isolates belonging to seven bacterial genera were collected at 20 (57%) different sampling occasions. The sinks located in the remaining rooms (no disinfection treatment) showed the broadest range of bacterial species and an even higher proportion of bacterial growth (see Figure 1).
Fig. 1. The number of bacterial isolates and type of bacteria are shown per room. Each room has two sinks. The percentage of bacterial growth in correlation with the number of sampling occasions are shown above each room.
The distribution of bacteria in the water traps of the sinks in the patient rooms and the bathrooms varied. In room 1, the majority (91%) of the bacteria were sampled from the bathroom. The corresponding figures for rooms 2-7 were 46%, 39%, 42%, 51%, 54%, and 70%, respectively.
The occupancy of the rooms in the burn centre differed. The room with the highest level of occupancy was room 2. It was occupied by four patients during 29 of the study weeks (83%). In contrast, room 7 was only occupied during two weeks (6%) and by two patients. This was the lowest level of occupancy. The remaining rooms were occupied as follows: room 1 by four patients during 23 weeks (66%), room 3 by seven patients during 18 weeks (51%), room 4 by five patients during 22 weeks (63%), room 5 by eight patients during 17 weeks (49%), and room 6 by six patients during 25 weeks (71%). The bacterial growth in the sinks, in correlation with patient occupancy per room, is shown in Figure 2. In all rooms but room 1, an increased accumulation of bacteria was observed when a patient was admitted to the room.
Fig. 2. The percentage of bacterial growth in correlation with patient occupancy are shown for each room.
The antibiotic susceptibility testing revealed a multidrug-resistant E. coli strain sampled from sinks in room 4. It was ESBL-producing, resistant to cefotaxim, ceftazidim, cefepim, piperacillin-tazobactam, nalidixic acid, ciprofloxacin, tobramycin and trimethoprim-sulfamethoxazole, and had been brought into the unit by the patient staying in the room. It was detected in the sinks for a longer time period. After the patient was discharged, the two sinks were treated with boiling water, and no new patient was admitted until the cultures were negative. P. aeruginosa isolates with resistance to ceftazidim, imipenem and meropenem were observed on different sampling occasions from sinks in rooms 1, 2 and 4. The remaining isolates showed no deviant resistance patterns.
Patient samples
A total of 36 patients were admitted to the Burn Centre during the study period. The duration of the stay varied depending on the severity of the burn injuries, e.g. room 2 was occupied by the same patient during 20 weeks before relocation, whereas another patient stayed for less than one week in room 5.
Culture samples collected from the patients showed the following growth of Gram-negatives: P. aeruginosa (n=31), E. cloacae (n=28), E. coli (n=11), Klebsiella spp. (n=7), Proteus spp. (n=6), S. maltophilia (n=6), Acinetobacter spp. (n=3), Serratia marcescens (n=2), Citrobacter freundii (n=2), Morganella morganii (n=1), Enterobacter amnigenus (n=1) and Moraxella catarrhalis (n=1). The growth of Gram-positive bacteria consisted of S. aureus (n=274), coagulase-negative staphylococci (n=88), Enterococcus spp (n=89), Streptococcus spp (n=37) and Bacillus spp (n=13).
Multidrug-resistant bacteria isolated from patients included seven samples of MRSA isolated from two patients admitted to room 6 on different occasions (unrelated strains and never found in any sink), and the multidrug-resistant E. coli found in the sinks in room 4. It was isolated from the patient at admittance. A P. aeruginosa strain with resistance to ceftazidim, ciprofloxacin, imipenem, meropenem and piperacillin/tazobactam was isolated at several different sampling occasions from a patient admitted to room 2. This patient was in addition colonized by an E. cloacae strain resistant to ceftazidim, cefotaxim and piperacillin/tazobactam.
WGS results
A total of 24 isolates were subjected to WGS: E. cloacae complex (n=8), P. aeruginosa (n=8), E. coli (n=4) and S. maltophilia (n=4). Two of the P. aeruginosa genomes were used twice, i.e. they were not only included when comparing sink-patient genomes but also when comparing patient-patient genomes when there was no growth of the bacterium in the sinks.
The samples obtained an average sequencing depth of 64 x. One cluster was recognized with MLST and whole genome-wide phylogenetic analysis. The cluster contained two isolates of E. coli, one sampled from a patient placed in room 6 and the other was an environmental sample collected from one of the sinks in the same room one week earlier. The isolates had a difference of a single SNP and were identified as sequence type (ST) 625. The remaining isolates belonged all to unique clones.
Isolates identified with MALDI-TOF mass spectrometry as E. cloacae complex constituted a special problem. In half of the cases, the genomes that were compared did not belong to the same species. Species within the complex identified with WGS included Enterobacter roggenkampii, Enterobacter hormaechei and Enterobacter ludwigii.
The E. cloacae complex is known for its ability to harbour the plasmid-mediated sil operon, a gene cluster encoding efflux pumps, a silver-binding protein and regulatory genes that confers resistance to silver [20]. Silver products are often used in burn centres and could therefore select for this bacterial complex, which was a relatively frequent finding in both sink and patient samples. The genomes of isolates belonging to the E. cloacae complex were therefore screened for the sil operon. Six out of eight isolates (75%) carried the full operon.
Discussion
There has been a clear increase of sink-associated outbreaks caused by Gram-negative bacteria in late years [21-26]. In the present study, it was investigated if stainless steel sinks, in which both the bowl and the water-trap were self-disinfected three times per 24 h, could reduce the bacterial load and thereby the risk of transmission. Furthermore, two conventional sinks were treated weekly with boiling water as an easy and cheaper alternative. The results showed that both alternatives reduced the bacterial load of the sinks compared with no disinfection at all, but the self-disinfecting sinks were significantly more efficient. This is in accord with other studies in which self-disinfecting sink drains have been used [27, 28].
The self-disinfecting sinks in room 1 had the overall lowest frequency of bacterial growth and the fewest number of species isolated during the entire study period. In contrast to all the other rooms, there was no correlation between patient occupancy and the bacterial growth in the sinks; the bacterial load remained low or was zero despite a suboptimal use of the sinks during patient care. Although the routine to initiate a self-disinfecting cycle every 8 h did not eliminate all bacterial growth, it showed that it could be radically decreased. It is quite possible that the bacterial growth would have been further reduced if the self-disinfecting cycle had been started every time the sink was contaminated. The health care personnel at the centre found, however, this instruction too complicated and time-consuming, why it was changed to every 8 h.
Treatment with boiling water was a simple and functioning alternative that kept the bacterial load at a relatively low level in the sinks located in room 4. As shown in a study from 2021, the initial concentration of bacteria in the drain is back within approximately a week [29]. Thus, this alternative needs to be carried out at least once weekly and continuously to avoid re-occurrence of growth. In addition, the procedure involves an extra workload for the personnel, and there is always a risk of contracting burn injuries while handling the boiling water. It was, however, chosen over chlorine, the traditional disinfectant for hospital sinks, since it has been shown to be 100 to 1000 times more effective in reducing pathogens, it does not smell, it is environmentally friendly and fairly inexpensive [29]. Replacement of contaminated sinks has been shown to reduce the infection rates in ICUs [30, 31], but bacteria may not only reside in the water-trap. They can also be found further down in the drainage system. As a result, bacteria can reappear despite a complete change of sinks [24]. Self-disinfecting sinks are therefore a better and the most long-term and solution to the problem.
The whole genome-wide phylogenetic analysis identified one cluster among the 24 patient and environmental samples that were randomly chosen. The cluster consisted of two E. coli isolates belonging to ST625, a clone associated with extra-intestinal infections [32]. The sink isolate was collected one week earlier than the patient isolate, indicating that the sink was the likely source of the bacterium that colonized the patient’s burn wounds. This is, to our knowledge, the first time this type of event has been observed prospectively in a clinical setting. The exact route for the transmission is, however, not clear. Few studies deal with the exact mechanism of transmission from a sink to a patient. In a recent study, mobilization of bacteria from biofilms in water-traps of sinks to the surrounding environment was demonstrated by using green fluorescent-expressing E. coli [12]. This is a possible transmission route for the E. coli in room 6.
Additional transmissions may also have occurred in this and other rooms, but the low number of isolates investigated and the fact that only a single colony was used when preparing the DNA limited the chances of detecting them. Interestingly, in the five cases where a patient was colonized by the same Gram-negative species as the former patient, and the sinks lacked growth of the species of interest, no transmission was observed. However, the number of colonies/isolates investigated can once again have been too low.
There were few multidrug-resistant isolates in the present study, but resistance does not always come in the form of antibiotic resistance. The isolation frequency of E. cloacae complex was relatively high among the Gram-negatives. Only one isolate was resistant to more broad-spectrum beta-lactams, whereas the carriage rate of the sil operon was quite high, at 75%. In an earlier study [20], 48% of invasive E. cloacae isolates harboured sil genes. These findings suggest that the use of silver products rather than antibiotics could have selected for this complex, but if the genes were expressed or not was never tested.
Although the main focus in this study was on Gram-negative bacteria, it was striking how few Gram-positive bacteria were isolated from the sinks compared with from the patients. For instance, only a single S. aureus isolate was recorded from the sinks. The corresponding figure from patients was 274, indicating that water traps offer mainly an environment that promotes growth of Gram-negative bacteria, and of S. maltophilia and P. aeruginosa in particular. However, even if S. aureus did not thrive in the water traps, it may survive, together with other Gram-positive bacteria and Acinetobacter spp., in the wash bowl. To reduce the risk of dissemination from this part of the sink, the wash bowl was also decontaminated during the disinfection process.
In conclusion, the results showed prospectively that sinks can serve as a reservoir of Gram-negative bacteria, and that self-disinfecting sinks can reduce the bacterial load in the sinks and thereby also the risk of bacterial transmission. Installing self-disinfecting sinks in ICUs is therefore an important measure in preventing nosocomial infection among critically ill and vulnerable patients. A less expensive but not as efficient solution can be to disinfect sinks with boiling water once weekly.
Funding
This study was financially supported by the National Association Heart-Lung and ALF-funds.
Acknowledgements
We thank the staff at the Burn Centre at Linköping University Hospital, Sweden, for their excellent help during the study period.
Author Contributions
Conceptualization, visualization, and data interpretation: M.G., Å.M.; Methodology and Analysis: M.G., T.F., J.W., Å.M.; Funding and Resources: Å.M.; Writing – original and revised draft preparation: M.G., J.W., Å.M. All authors have read and agreed to the published version of the manuscript.
Conflicts of Interest
The self-disinfecting sink was invented by Micropharmics AB (owned by Å.M.) and developed together with Tunerlux AB. The funder had no role in study design, data collection and interpretation, or the decision to submit the work for publication.
References
- Coppola, N.; Maraolo, A.E.; Onorato, L.; Scotto, R.; Calo, F.; Atripaldi, L.; Borrelli, A.; Corcione, A.; De Cristofaro, M.G.; Durante-Mangoni, E.; Filippelli, A.; Franci, G.; Galdo, M.; Guglielmi, G.; Pagliano, P.; Perrella, A.; Piazza, O.; Picardi, M.; Punzi, R.; Trama, U.; Gentile, I. Epidemiology, Mechanisms of Resistance and Treatment Algorithm for Infections Due to Carbapenem-Resistant Gram-Negative Bacteria: An Expert Panel Opinion. Antibiotics (Basel). 2022, 11. doi: 10.3390/antibiotics11091263.
- Wagner, T.M.; Howden, B.P.; Sundsfjord, A.; Hegstad, K. Transiently silent acquired antimicrobial resistance: an emerging challenge in susceptibility testing. J Antimicrob Chemother. 2023, 78, 586-98. doi: 10.1093/jac/dkad024.
- Serra-Burriel, M.; Keys, M.; Campillo-Artero, C.; Agodi, A.; Barchitta, M.; Gikas, A.; Palos, C.; Lopez-Casasnovas, G. Impact of multi-drug resistant bacteria on economic and clinical outcomes of healthcare-associated infections in adults: Systematic review and meta-analysis. PLoS One. 2020, 15, e0227139. doi: 10.1371/journal.pone.0227139.
- Antimicrobial Resistance C. Global burden of bacterial antimicrobial resistance in 2019: a systematic analysis. Lancet. 2022, 399, 629-55. doi: 10.1016/S0140-6736(21)02724-0.
- MacVane, S.H. Antimicrobial Resistance in the Intensive Care Unit: A Focus on Gram-Negative Bacterial Infections. J Intensive Care Med. 2017, 32, 25-37. doi: 10.1177/0885066615619895.
- Sader, H.S.; Mendes, R.E.; Streit, J.M.; Carvalhaes, C.G.; Castanheira, M. Antimicrobial susceptibility of Gram-negative bacteria from intensive care unit and non-intensive care unit patients from United States hospitals (2018-2020). Diagn Microbiol Infect Dis. 2022, 102, 115557. doi: 10.1016/j.diagmicrobio.2021.115557.
- Decker, B.K.; Palmore, T.N. The role of water in healthcare-associated infections. Curr Opin Infect Dis. 2013, 26, 345-51. doi: 10.1097/QCO.0b013e3283630adf.
- Ladhani, H.A.; Yowler, C.J.; Claridge, J.A. Burn Wound Colonization, Infection, and Sepsis. Surg Infect (Larchmt). 2021, 22, 44-8. doi: 10.1089/sur.2020.346.
- Yin, Z.; Beiwen, W.; Zhenzhu, M.; Erzhen, C.; Qin, Z.; Yi, D. Characteristics of bloodstream infection and initial antibiotic use in critically ill burn patients and their impact on patient prognosis. Sci Rep. 2022, 12, 20105. doi: 10.1038/s41598-022-24492-z.
- Haider, M.H.; McHugh, T.D.; Roulston, K.; Arruda, L.B.; Sadouki, Z.; Riaz, S. Detection of carbapenemases bla(OXA48)-bla(KPC)-bla(NDM)-bla(VIM) and extended-spectrum-beta-lactamase bla(OXA1)-bla(SHV)-bla(TEM) genes in Gram-negative bacterial isolates from ICU burns patients. Ann Clin Microbiol Antimicrob. 2022, 21, 18. doi: 10.1186/s12941-022-00510-w.
- Lima, W.G.; Silva Alves, G.C.; Sanches, C.; Antunes Fernandes, S.O.; de Paiva, M.C. Carbapenem-resistant Acinetobacter baumannii in patients with burn injury: A systematic review and meta-analysis. Burns. 2019, 45, 1495-508. doi: 10.1016/j.burns.2019.07.006.
- Kotay, S.; Chai, W.; Guilford, W.; Barry, K.; Mathers, A.J. Spread from the Sink to the Patient: In Situ Study Using Green Fluorescent Protein (GFP)-Expressing Escherichia coli To Model Bacterial Dispersion from Hand-Washing Sink-Trap Reservoirs. Appl Environ Microbiol. 2017, 83. doi: 10.1128/AEM.03327-16.
- Valentin, A.S.; Santos, S.D.; Goube, F.; Gimenes, R.; Decalonne, M.; Mereghetti, L.; Daniau, C.; van der Mee-Marquet, N. A prospective multicentre surveillance study to investigate the risk associated with contaminated sinks in the intensive care unit. Clin Microbiol Infect. 2021, 27, 1347 e9- e14. doi: 10.1016/j.cmi.2021.02.018.
- Kearney, A.; Boyle, M.A.; Curley, G.F.; Humphreys, H. Preventing infections caused by carbapenemase-producing bacteria in the intensive care unit - Think about the sink. J Crit Care. 2021, 66, 52-9. doi: 10.1016/j.jcrc.2021.07.023.
- Loveday, H.P.; Wilson, J.A.; Kerr, K.; Pitchers, R.; Walker, J.T.; Browne, J. Association between healthcare water systems and Pseudomonas aeruginosa infections: a rapid systematic review. J Hosp Infect. 2014, 86, 7-15. doi: 10.1016/j.jhin.2013.09.010.
- Volling, C.; Ahangari, N.; Bartoszko, J.J.; Coleman, B.L.; Garcia-Jeldes, F.; Jamal, A.J.; Johnstone, J.; Kandel, C.; Kohler, P.; Maltezou, H.; Maze Dit Mieusement, L.; McKenzie, N.; Mertz, D.; Monod, A.; Saeed, S.; Shea, B.; Stuart, R.; Thomas, S.; Uleryk, E.; McGeer, A. Are Sink Drainage Systems a Reservoir for Hospital-Acquired Gammaproteobacteria Colonization and Infection? A Systematic Review. Open Forum Infect Dis. 2021, 8, ofaa590. doi: 10.1093/ofid/ofaa590.
- Bayuo, J.; Agbenorku, P. Coping strategies among nurses in the Burn Intensive Care Unit: A qualitative study. Burns Open. 2018, 2, 47-52. doi: 10.1016/J.BURNSO.2017.10.004.
- Schurch, A.C.; Arredondo-Alonso, S.; Willems, R.J.L.; Goering, R.V. Whole genome sequencing options for bacterial strain typing and epidemiologic analysis based on single nucleotide polymorphism versus gene-by-gene-based approaches. Clin Microbiol Infect. 2018, 24, 350-4. doi: 10.1016/j.cmi.2017.12.016.
- Pightling, A.W.; Pettengill, J.B.; Luo, Y.; Baugher, J.D.; Rand, H.; Strain, E. Interpreting Whole-Genome Sequence Analyses of Foodborne Bacteria for Regulatory Applications and Outbreak Investigations. Front Microbiol. 2018, 9, 1482. doi: 10.3389/fmicb.2018.01482.
- Sutterlin, S.; Dahlo, M.; Tellgren-Roth, C.; Schaal, W.; Melhus, A. High frequency of silver resistance genes in invasive isolates of Enterobacter and Klebsiella species. J Hosp Infect. 2017, 96, 256-61. doi: 10.1016/j.jhin.2017.04.017.
- Gideskog, M.; Welander, J.; Melhus, A. Cluster of S. maltophilia among patients with respiratory tract infections at an intensive care unit. Infect Prev Pract. 2020, 2, 100097. doi: 10.1016/j.infpip.2020.100097.
- Jung, J.; Choi, H.S.; Lee, J.Y.; Ryu, S.H.; Kim, S.K.; Hong, M.J.; Kwak, S.H.; Kim, H.J.; Lee, M.S.; Sung, H.; Kim, M.N.; Kim, S.H. Outbreak of carbapenemase-producing Enterobacteriaceae associated with a contaminated water dispenser and sink drains in the cardiology units of a Korean hospital. J Hosp Infect. 2020,104, 476-83. doi: 10.1016/j.jhin.2019.11.015.
- Catho, G.; Martischang, R.; Boroli, F.; Chraiti, M.N.; Martin, Y.; Koyluk Tomsuk, Z.; Renzi, G.; Schrenzel, J.; Pugin, J.; Nordmann, P.; Blanc, D.S.; Harbarth, S. Outbreak of Pseudomonas aeruginosa producing VIM carbapenemase in an intensive care unit and its termination by implementation of waterless patient care. Crit Care. 2021, 25, 301. doi: 10.1186/s13054-021-03726-y.
- Stjarne Aspelund, A.; Sjostrom, K.; Olsson Liljequist, B.; Morgelin, M.; Melander, E.; Pahlman, L.I. Acetic acid as a decontamination method for sink drains in a nosocomial outbreak of metallo-beta-lactamase-producing Pseudomonas aeruginosa. J Hosp Infect. 2016, 94, 13-20. doi: 10.1016/j.jhin.2016.05.009.
- Rehou, S.; Rotman, S.; Avaness, M.; Salt, N.; Jeschke, M.G.; Shahrokhi, S. Outbreak of carbapenemase-producing Enterobacteriaceae in a regional burn centre. J Burn Care Res. 2022, 43, 1203-6. doi: 10.1093/jbcr/irac067.
- Salm, F.; Deja, M.; Gastmeier, P.; Kola, A.; Hansen, S.; Behnke, M.; Gruhl, D.; Leistner, R. Prolonged outbreak of clonal MDR Pseudomonas aeruginosa on an intensive care unit: contaminated sinks and contamination of ultra-filtrate bags as possible route of transmission? Antimicrob Resist Infect Control. 2016, 5, 53. doi: 10.1186/s13756-016-0157-9.
- de Jonge, E.; de Boer, M.G.J.; van Essen, E.H.R.; Dogterom-Ballering, H.C.M.; Veldkamp, K.E. Effects of a disinfection device on colonization of sink drains and patients during a prolonged outbreak of multidrug-resistant Pseudomonas aeruginosa in an intensive care unit. J Hosp Infect. 2019, 102, 70-4. doi: 10.1016/j.jhin.2019.01.003.
- Fusch, C.; Pogorzelski, D.; Main, C.; Meyer, C.L.; El Helou, S.; Mertz, D. Self-disinfecting sink drains reduce the Pseudomonas aeruginosa bioburden in a neonatal intensive care unit. Acta Paediatr. 2015, 104, e344-9. doi: 10.1111/apa.13005.
- Diamond, F. Hot water might disinfect sinks better than chlorine. Infection control today. Available online: https://www.infectioncontroltoday.com/view/hot-water-might-disinfect-sinks-better-than-chlorine (accessed on 15 November 2022).
- Hota, S.; Hirji, Z.; Stockton, K.; Lemieux, C.; Dedier, H.; Wolfaardt, G.; Gardam, M. Outbreak of multidrug-resistant Pseudomonas aeruginosa colonization and infection secondary to imperfect intensive care unit room design. Infect Control Hosp Epidemiol. 2009, 30, 25-33. doi: 10.1086/592700.
- Longtin, Y.; Troillet, N.; Touveneau, S.; Boillat, N.; Rimensberger, P.; Dharan, S.; Gervaix, A.; Pittet, D.; Harbarth, S. Pseudomonas aeruginosa outbreak in a pediatric intensive care unit linked to a humanitarian organization residential center. Pediatr Infect Dis J. 2010, 29, 233-7. doi: 10.1097/INF.0b013e3181bc24fb.
- Alonso, C.A.; Gonzalez-Barrio, D.; Ruiz-Fons, F.; Ruiz-Ripa, L.; Torres, C. High frequency of B2 phylogroup among non-clonally related fecal Escherichia coli isolates from wild boars, including the lineage ST131. FEMS Microbiol Ecol. 2017, 93. doi: 10.1093/femsec/fix016.
Reviewer 3 Report
The manuscript “The first Swedish outbreak with VIM-2-producing Pseudomonas aeruginosa was pro- 2 longed and probably due to contaminated hospital sinks” by Carl-Johan Fraenkel et al. described the first outbreak of 59 VIM-2-producing P. aeruginosa in Sweden, and the route of transmission was speculated. Although the authors put forwarded the sinks and other water sources in the hospital environment should be considered when facing the prolonged outbreaks. It needs more proofs to verify the route of transmission, as the author also denoted new cases occurred in a tertiary care hospital in the region in the abstract section. It is desirable to have quantitative laboratory evidence to prove that wet environment is more conducive to the survival of P. aeruginosa. The work is systematic and novel, the results may be useful for preventing the P. aeruginosa.
Author Response
Reviewer 3
Reviewer’s comment: Although the authors put forwarded the sinks and other water sources in the hospital environment should be considered when facing the prolonged outbreaks. It needs more proofs to verify the route of transmission, as the author also denoted new cases occurred in a tertiary care hospital in the region in the abstract section. It is desirable to have quantitative laboratory evidence to prove that wet environment is more conducive to the survival of P. aeruginosa.
Authors' response: This comment is very similar to the one we got when we first submitted our manuscript many years ago, but to give a direct proof of a transmission of VIM-2-producing P. aeruginosa from a sink to a patient is a tall order. We have only one or two cases per year in the whole country. What we have been able to carry out is, however, a prospective study where we explored if the bacterial load in sinks could be reduced by using self-disinfecting sinks. The design of the study made it possible for us to show that Gram-negative bacteria were transmitted from a sink to a patient (see below). That P. aeruginosa survives better in a wet environment than many other bacteria and poorly on dry surfaces has been shown by other research groups. We have added few words and a references dealing with this in the Discussion. We have also included a few words in the Discussion about the tertiary care hospital in Lund and its role in generating patients over the years. In contrast to all other hospitals in this region, Lund has still growth of the VIM-2-producing P. aeruginosa strain in some of their sinks and plumbing system. They are therefore still treating several sinks with acetic acid.
Source Control of Gram-negative Bacteria by using Self-Disinfecting Sinks in a Swedish Burn Centre
Maria Gideskog1, Tina Falkeborn2, Jenny Welander2, Åsa Melhus3
1Department of Communicable Disease and Infection Control, and Department of Biomedical and Clinical Sciences, Linköping University, Linköping
2Department of Clinical Microbiology, and Department of Biomedical and Clinical Sciences, Linköping University, Linköping
3Department of Medical Sciences, Section of Clinical Microbiology, Uppsala University, Uppsala, Sweden
Corresponding author: Maria Gideskog, Department of Communicable Disease and Infection Control, and Department of Biomedical and Clinical Sciences, Linköping University, Gasverksgränd 2, SE-581 85 Linköping, Sweden. Telephone: +46 (0)101036851. E-mail: maria.gideskog@regionostergotland.se.
Abstract
Several retrospective studies have identified hospital sinks as a reservoir of Gram-negative bacteria. The aim of this study was to investigate prospectively the bacterial transmission from sinks to patients and if self-disinfecting sinks could reduce this risk. Samples were collected weekly from sinks (self-disinfecting, treated with boiling water, not treated) and patients in the Burn Centre at Linköping University Hospital, Sweden. The antibiotic susceptibility of Gram-negative isolates was tested, and eight randomly chosen patient isolates and their connected sink isolates were subjected to whole genome sequencing (WGS). Of 489 sink samples, 232 (47%) showed growth. The most frequent findings were Stenotrophomonas maltophilia (n=130), Pseudomonas aeruginosa (n=128) and Acinetobacter spp. (n=55). Bacterial growth was observed in 20% of the samplings from the self-disinfecting sinks and in 57% from the sinks treated with boiling water (P=0.0029). WGS recognized one transmission of Escherichia coli sampled from an untreated sink to a patient admitted to the same room. In conclusion, the results showed that sinks can serve as a reservoir of Gram-negative bacteria and that self-disinfecting sinks can reduce the transmission risk. Installing self-disinfecting sinks in intensive care units is an important measure in preventing nosocomial infection among critically ill patients.
Keywords: sink; water-trap; bacterial transmission; self-disinfecting sink; infection control; Pseudomonas aeruginosa; Stenotrophomonas maltophilia; Acinetobacter
Introduction
Healthcare-related infections caused by multidrug-resistant Gram-negative bacteria are medically challenging. Few treatment options are usually available due to the wide and complex range of resistance mechanisms these bacteria carry [1, 2], and, as a consequence, they are associated with an increased financial burden, prolonged hospital stays and an increased mortality [3, 4].
In intensive care units (ICUs), the clinical impact of opportunistic Gram-negative bacteria with multidrug-resistance, such as Acinetobacter baumanii, Enterobacter cloacae, Klebsiella pneumonia, Pseudomonas aeruginosa and Stenotrophomonas maltophilia, is increasing. Medical conditions associated with these bacteria range from colonization of the respiratory and urinary tract to deep and disseminated infections [5-7].
In this context, burn patients represent an especially difficult cohort. A loss of a functioning skin barrier in the form of a third-degree burn, often combined with an inhalation injury and endotracheal intubation, entails a dysfunctional immune system and a high vulnerability to colonization of Gram-negative bacteria. Infections are frequent, and they can lead to everything from melting skin grafts to septic shock and death [8]. The more severe or larger the burn injury is, the more likely is it that an infection will ensue. To prevent serious complications, it is essential to have a proactive approach and treat the infection as early and efficient as possible. Cultures are therefore often regularly performed and repeated courses of antibiotics prescribed. The high selective pressure favours multidrug-resistance, and common bacterial findings are Acinetobacter spp., K. pneumoniae and P. aeruginosa [9-11].
With few exceptions, Gram-negative bacteria are sensitive to desiccation. They are therefore typically found in moist environments, e.g. sinks and their drainage systems. The water traps of sinks constitute a relatively protected environment, which favours the growth of bacteria and production of biofilms [12-14]. Once biofilms have been established, disinfectants cannot fully eradicate them [15]. Through splash water and aerosols, bacteria can be mobilized and transmitted from the sinks to patients. Sinks have been identified as a potential source of infections and outbreaks in ICUs in several reports, but their clinical importance has to some extent been questioned due to the lack of prospective studies available [16].
A burn centre is a complex and stressful care environment [17]. Operations are usually performed in the patient room to avoid moving the patient, and the patient may stay for several months. Thus, the sinks located in the patient rooms are frequently used for other purposes than hand washing. Gram-negative bacteria therefore tend to accumulate in the sinks and their drainage systems. To explore the extent of sink contamination, samples from water traps in sinks at the Burn Centre at Linköping University Hospital, Sweden, were cultured the summer 2018. Growth of clinically relevant Gram-negative bacteria was recorded in all sinks placed in patient rooms and the associated bathrooms. Furthermore, several of the identified species were also observed in blood and wound samples from admitted patients.
The aim of this study was to investigate if it would be possible to reduce the load of Gram-negative bacteria in sinks, and thereby also indirectly the risk of nosocomial infections in a burn centre, by installing newly developed self-disinfecting sinks. The design of the study made it also possible to explore prospectively the transmission of Gram-negative bacteria from sinks to patients.
Materials and Methods
Settings
The Burn Centre at Linköping University Hospital, Östergötland County, Sweden, is one of two units for national highly specialised care of severely burned patients in Sweden. Approximately 100 patients are admitted each year. The catch area is nationwide, but the majority of the patients are referred from the south of Sweden. The unit offers a total of seven single-bed rooms, of which four (rooms 1-4) are equipped for intensive care with a high level of medical monitoring and access to respiratory care. There are two sinks per room, one located in the patient room and the other in the bathroom. The sinks are used for hand washing, cleaning of various medical devices and in the direct patient care.
Since the study material only comprised bacterial isolates and no changes were made in well-established clinical routines, no ethical approval was sought.
Self-disinfecting sinks
The self-disinfecting stainless steel sink (Dissinkfect®, Micropharmics AB and Tunerlux AB, Uppsala, Sweden) used in this study has a built-in heating supply, which heat the wash bowl to 75°C and the water trap to 100°C. It tolerates quick temperature changes and commonly used cleansing or disinfecting agents. By pressing a button placed on the side of the sink for four seconds, the disinfection-process starts and a green LED-indicator shines during the whole 15 min process. It can be stopped at any time, and its length and temperatures can be adjusted according to the requirements. During the study period, self-disinfection was initiated once per each work shift, i.e. three times per 24 h.
Two self-disinfecting sinks, one located in the patient room and one in the bathroom, were installed in room 1. This was the intensive care room most frequently occupied by patients prior to the study. Another intensive care room (room 4) was selected as a comparator, and the sinks in this room were treated with boiling water (3 L each) once a week during the entire study period. The remaining sinks at the centre acted as controls and were not disinfected at any time. All sinks were cleaned daily.
Environmental cultures
To explore the growth of different bacteria in the water trap of sinks over time, environmental samples were collected with ESwabs (Copan Diagnostics Inc. Murrieta, CA, USA) from all 14 patient-associated sinks in the Burn Centre. The sampling started directly after the installation of the self-disinfecting sinks in September 2019 and continued on a weekly basis until April 2020, a total of 35 weeks. Records were kept concerning patient occupancy of each room upon sampling.
The samples were sent to the Department of Clinical Microbiology, Linköping University Hospital, and inoculated onto three different types of media: blood agar, hematin agar and chromogenic urinary tract infection (UTI) agar (Thermo Fisher Scientific, Waltham, MA, USA). Discs (Thermo Fisher Scientific, Waltham, MA, USA) with imipenem (10 µg), trimethoprim-sulfamethoxazole (1.25-23.75 µg), and linezolid (10 µg) was placed on the plates, respectively. The plates were incubated at 35°C for approximately 48 h. Bacteria were identified to the species level with a MALDI Biotyper 3.0 (Bruker Corporation, Karlsruhe, Germany).
Patients
All patients admitted to the Burn Centre during the study period were cultured once a week and upon any clinical sign of infection, according to the routines of the unit. ESwabs (Copan Diagnostics Inc.) were used when sampling from burn wounds. The samples were inoculated onto four different types of media: hematin agar, chromogenic UTI agar, streptococcus agar and chromogenic Staphylococcus aureus (CSA) agar (Thermo Fisher Scientific, Waltham, MA, USA) and incubated at 35°C for approximately 48 h. All Gram-negative isolates were frozen at -70℃ to allow for future genetic analyses. Gram-positive isolates were only frozen if it was indicated in the quality manual of the laboratory. During the patient’s stay, it was recorded in which room the patient was placed. Relocation of patients was avoided, unless a patient was moved from a room equipped for intensive care to a regular room as a result of treatment progress.
Antibiotic susceptibility testing
The antibiotic susceptibility was tested with the disc diffusion method according to the recommendations of EUCAST (www.eucast.org). For environmental Gram-negative bacteria it included cefotaxim, ceftazidim, cefepim, piperacillin-tazobactam, imipenem, meropenem, nalidixic acid, ciprofloxacin, tobramycin and trimethoprim-sulfamethoxazole. For P. aeruginosa the susceptibility testing was limited to ceftazidim, imipenem and meropenem, and for S. maltophilia to trimethoprim-sulfamethoxazole.
Patient isolates were tested when judged clinically relevant and against antibiotics recommended for each species. S. aureus resistant to cefoxitin were further analysed with polymerase chain reaction (PCR) to determine carriage of the nuc gene and mecA gene. All methicillin-resistant S. aureus (MRSA) isolates were subjects for whole genome sequencing (WGS).
An isolate was considered multidrug-resistant if it was resistant to at least three classes of antibiotics, P. aeruginosa and S. maltophilia exempted. Enterobacterales with cefotaxim/ceftazidim/cefepim/meropenem susceptibility patterns raising concerns about extended spectrum beta-lactamase (ESBL)-production were further evaluated fenotypically and/or genotypically, in accordance with local guidelines.
WGS
Eight randomly chosen isolates from the same number of patients, and isolates from water-traps of sinks that belonged to the same species and were connected in space and time to each patient, were subjected to WGS. Furthermore, there were rooms in which a patient was colonized by the same species as the former patient, but there was no growth of this species in the sinks. Five of these patients’ isolates were randomly selected for sequencing, together with five isolates from former patients. The patients had stayed in rooms 1, 2, 4, 5, and 6.
DNA was prepared from 1µL from a single colony of each isolate, using EZ1 DNA Tissue Kit (Qiagen, Germantown, MD, USA), with an included pre-heating step at 95°C and shaking at 350 rpm. Twenty ng of DNA was used for library preparation, using QIAseq FX DNA Library Kit (Qiagen, Germantown, MD, USA) with 8 min of fragmentation time. DNA libraries were sequenced on the MiSeq platform (Illumina, San Diego, CA, USA) with 2 x 300 bp paired-end reads.
Data analysis was performed in CLC Genomics Workbench v. 10.1.1 with the Microbial Genomics Module v. 2.5.1 (Qiagen, Germantown, MD, USA). Multilocus sequence typing (MLST) analysis was performed using the PubMLST (pubmlst.org) scheme for each randomly chosen bacterial species. Read mapping and variant calling were performed against the different reference genomes with NCBI accession numbers NC_008253 (Escherichia coli), NC_018405 (E. cloacae), NC_017548 (P. aeruginosa) and NC_010943 (S. maltophilia), with the following thresholds to call a variant: depth of coverage ≥ 20x, frequency ≥ 90% and Phred score ≥ 20. A quality filter was then applied that retained variants with a sequencing depth of ≥ 20x in all samples and a distance ≥ 10 bp to the next variant, and the resulting variants were used to create single nucleotide polymorphism (SNP) trees and calculate genetic distances between samples. Previous studies suggest that isolates of E. coli and P. aeruginosa with a distance of ≤10 SNPs and ≤37 SNPs, respectively, are likely to belong to the same clone [18]. So far, no studies have suggested SNP thresholds for E. cloacae or S. maltophilia, but an SNP distance of <21 has been considered to support that two bacterial isolates in general have arisen from the same source [19].
Statistical analysis
Fischer’s exact test was used when comparing the culture results from the three groups of sinks (self-disinfecting, treated with boiling water, not treated). A P-value of ≤0.05 was considered statistically significant.
Results
Environmental samples
A total of 489 samples were collected from the water-traps of the sinks during the study period. Of these, 232 samples (47%) showed growth of one or more bacterial species. The three most frequent Gram-negative bacteria were S. maltophilia (n=130), P. aeruginosa (n=128) and Acinetobacter spp (n=55). For more details, see Figure 1. The growth of Gram-positive bacteria consisted mostly of skin flora: coagulase-negative staphylococci (n=24), S. aureus (n=1) and Enterococcus faecalis (n=6).
Bacterial growth in one or both of the self-disinfecting sinks located in room 1 was observed at seven (20%) different sampling occasions. The bacterial load in these thinks were significantly lower than in those treated with boiling water once a week (P=0.0029) and those that were not treated at all (P=< 0.00001). The total number of Gram-negative isolates was eleven and consisted of Acinetobacter spp. (n=5), S. maltophilia (n=4) and P. aeruginosa (n=2).
In the sinks treated with boiling water in room 4, 57 Gram-negative bacterial isolates belonging to seven bacterial genera were collected at 20 (57%) different sampling occasions. The sinks located in the remaining rooms (no disinfection treatment) showed the broadest range of bacterial species and an even higher proportion of bacterial growth (see Figure 1).
Fig. 1. The number of bacterial isolates and type of bacteria are shown per room. Each room has two sinks. The percentage of bacterial growth in correlation with the number of sampling occasions are shown above each room.
The distribution of bacteria in the water traps of the sinks in the patient rooms and the bathrooms varied. In room 1, the majority (91%) of the bacteria were sampled from the bathroom. The corresponding figures for rooms 2-7 were 46%, 39%, 42%, 51%, 54%, and 70%, respectively.
The occupancy of the rooms in the burn centre differed. The room with the highest level of occupancy was room 2. It was occupied by four patients during 29 of the study weeks (83%). In contrast, room 7 was only occupied during two weeks (6%) and by two patients. This was the lowest level of occupancy. The remaining rooms were occupied as follows: room 1 by four patients during 23 weeks (66%), room 3 by seven patients during 18 weeks (51%), room 4 by five patients during 22 weeks (63%), room 5 by eight patients during 17 weeks (49%), and room 6 by six patients during 25 weeks (71%). The bacterial growth in the sinks, in correlation with patient occupancy per room, is shown in Figure 2. In all rooms but room 1, an increased accumulation of bacteria was observed when a patient was admitted to the room.
Fig. 2. The percentage of bacterial growth in correlation with patient occupancy are shown for each room.
The antibiotic susceptibility testing revealed a multidrug-resistant E. coli strain sampled from sinks in room 4. It was ESBL-producing, resistant to cefotaxim, ceftazidim, cefepim, piperacillin-tazobactam, nalidixic acid, ciprofloxacin, tobramycin and trimethoprim-sulfamethoxazole, and had been brought into the unit by the patient staying in the room. It was detected in the sinks for a longer time period. After the patient was discharged, the two sinks were treated with boiling water, and no new patient was admitted until the cultures were negative. P. aeruginosa isolates with resistance to ceftazidim, imipenem and meropenem were observed on different sampling occasions from sinks in rooms 1, 2 and 4. The remaining isolates showed no deviant resistance patterns.
Patient samples
A total of 36 patients were admitted to the Burn Centre during the study period. The duration of the stay varied depending on the severity of the burn injuries, e.g. room 2 was occupied by the same patient during 20 weeks before relocation, whereas another patient stayed for less than one week in room 5.
Culture samples collected from the patients showed the following growth of Gram-negatives: P. aeruginosa (n=31), E. cloacae (n=28), E. coli (n=11), Klebsiella spp. (n=7), Proteus spp. (n=6), S. maltophilia (n=6), Acinetobacter spp. (n=3), Serratia marcescens (n=2), Citrobacter freundii (n=2), Morganella morganii (n=1), Enterobacter amnigenus (n=1) and Moraxella catarrhalis (n=1). The growth of Gram-positive bacteria consisted of S. aureus (n=274), coagulase-negative staphylococci (n=88), Enterococcus spp (n=89), Streptococcus spp (n=37) and Bacillus spp (n=13).
Multidrug-resistant bacteria isolated from patients included seven samples of MRSA isolated from two patients admitted to room 6 on different occasions (unrelated strains and never found in any sink), and the multidrug-resistant E. coli found in the sinks in room 4. It was isolated from the patient at admittance. A P. aeruginosa strain with resistance to ceftazidim, ciprofloxacin, imipenem, meropenem and piperacillin/tazobactam was isolated at several different sampling occasions from a patient admitted to room 2. This patient was in addition colonized by an E. cloacae strain resistant to ceftazidim, cefotaxim and piperacillin/tazobactam.
WGS results
A total of 24 isolates were subjected to WGS: E. cloacae complex (n=8), P. aeruginosa (n=8), E. coli (n=4) and S. maltophilia (n=4). Two of the P. aeruginosa genomes were used twice, i.e. they were not only included when comparing sink-patient genomes but also when comparing patient-patient genomes when there was no growth of the bacterium in the sinks.
The samples obtained an average sequencing depth of 64 x. One cluster was recognized with MLST and whole genome-wide phylogenetic analysis. The cluster contained two isolates of E. coli, one sampled from a patient placed in room 6 and the other was an environmental sample collected from one of the sinks in the same room one week earlier. The isolates had a difference of a single SNP and were identified as sequence type (ST) 625. The remaining isolates belonged all to unique clones.
Isolates identified with MALDI-TOF mass spectrometry as E. cloacae complex constituted a special problem. In half of the cases, the genomes that were compared did not belong to the same species. Species within the complex identified with WGS included Enterobacter roggenkampii, Enterobacter hormaechei and Enterobacter ludwigii.
The E. cloacae complex is known for its ability to harbour the plasmid-mediated sil operon, a gene cluster encoding efflux pumps, a silver-binding protein and regulatory genes that confers resistance to silver [20]. Silver products are often used in burn centres and could therefore select for this bacterial complex, which was a relatively frequent finding in both sink and patient samples. The genomes of isolates belonging to the E. cloacae complex were therefore screened for the sil operon. Six out of eight isolates (75%) carried the full operon.
Discussion
There has been a clear increase of sink-associated outbreaks caused by Gram-negative bacteria in late years [21-26]. In the present study, it was investigated if stainless steel sinks, in which both the bowl and the water-trap were self-disinfected three times per 24 h, could reduce the bacterial load and thereby the risk of transmission. Furthermore, two conventional sinks were treated weekly with boiling water as an easy and cheaper alternative. The results showed that both alternatives reduced the bacterial load of the sinks compared with no disinfection at all, but the self-disinfecting sinks were significantly more efficient. This is in accord with other studies in which self-disinfecting sink drains have been used [27, 28].
The self-disinfecting sinks in room 1 had the overall lowest frequency of bacterial growth and the fewest number of species isolated during the entire study period. In contrast to all the other rooms, there was no correlation between patient occupancy and the bacterial growth in the sinks; the bacterial load remained low or was zero despite a suboptimal use of the sinks during patient care. Although the routine to initiate a self-disinfecting cycle every 8 h did not eliminate all bacterial growth, it showed that it could be radically decreased. It is quite possible that the bacterial growth would have been further reduced if the self-disinfecting cycle had been started every time the sink was contaminated. The health care personnel at the centre found, however, this instruction too complicated and time-consuming, why it was changed to every 8 h.
Treatment with boiling water was a simple and functioning alternative that kept the bacterial load at a relatively low level in the sinks located in room 4. As shown in a study from 2021, the initial concentration of bacteria in the drain is back within approximately a week [29]. Thus, this alternative needs to be carried out at least once weekly and continuously to avoid re-occurrence of growth. In addition, the procedure involves an extra workload for the personnel, and there is always a risk of contracting burn injuries while handling the boiling water. It was, however, chosen over chlorine, the traditional disinfectant for hospital sinks, since it has been shown to be 100 to 1000 times more effective in reducing pathogens, it does not smell, it is environmentally friendly and fairly inexpensive [29]. Replacement of contaminated sinks has been shown to reduce the infection rates in ICUs [30, 31], but bacteria may not only reside in the water-trap. They can also be found further down in the drainage system. As a result, bacteria can reappear despite a complete change of sinks [24]. Self-disinfecting sinks are therefore a better and the most long-term and solution to the problem.
The whole genome-wide phylogenetic analysis identified one cluster among the 24 patient and environmental samples that were randomly chosen. The cluster consisted of two E. coli isolates belonging to ST625, a clone associated with extra-intestinal infections [32]. The sink isolate was collected one week earlier than the patient isolate, indicating that the sink was the likely source of the bacterium that colonized the patient’s burn wounds. This is, to our knowledge, the first time this type of event has been observed prospectively in a clinical setting. The exact route for the transmission is, however, not clear. Few studies deal with the exact mechanism of transmission from a sink to a patient. In a recent study, mobilization of bacteria from biofilms in water-traps of sinks to the surrounding environment was demonstrated by using green fluorescent-expressing E. coli [12]. This is a possible transmission route for the E. coli in room 6.
Additional transmissions may also have occurred in this and other rooms, but the low number of isolates investigated and the fact that only a single colony was used when preparing the DNA limited the chances of detecting them. Interestingly, in the five cases where a patient was colonized by the same Gram-negative species as the former patient, and the sinks lacked growth of the species of interest, no transmission was observed. However, the number of colonies/isolates investigated can once again have been too low.
There were few multidrug-resistant isolates in the present study, but resistance does not always come in the form of antibiotic resistance. The isolation frequency of E. cloacae complex was relatively high among the Gram-negatives. Only one isolate was resistant to more broad-spectrum beta-lactams, whereas the carriage rate of the sil operon was quite high, at 75%. In an earlier study [20], 48% of invasive E. cloacae isolates harboured sil genes. These findings suggest that the use of silver products rather than antibiotics could have selected for this complex, but if the genes were expressed or not was never tested.
Although the main focus in this study was on Gram-negative bacteria, it was striking how few Gram-positive bacteria were isolated from the sinks compared with from the patients. For instance, only a single S. aureus isolate was recorded from the sinks. The corresponding figure from patients was 274, indicating that water traps offer mainly an environment that promotes growth of Gram-negative bacteria, and of S. maltophilia and P. aeruginosa in particular. However, even if S. aureus did not thrive in the water traps, it may survive, together with other Gram-positive bacteria and Acinetobacter spp., in the wash bowl. To reduce the risk of dissemination from this part of the sink, the wash bowl was also decontaminated during the disinfection process.
In conclusion, the results showed prospectively that sinks can serve as a reservoir of Gram-negative bacteria, and that self-disinfecting sinks can reduce the bacterial load in the sinks and thereby also the risk of bacterial transmission. Installing self-disinfecting sinks in ICUs is therefore an important measure in preventing nosocomial infection among critically ill and vulnerable patients. A less expensive but not as efficient solution can be to disinfect sinks with boiling water once weekly.
Funding
This study was financially supported by the National Association Heart-Lung and ALF-funds.
Acknowledgements
We thank the staff at the Burn Centre at Linköping University Hospital, Sweden, for their excellent help during the study period.
Author Contributions
Conceptualization, visualization, and data interpretation: M.G., Å.M.; Methodology and Analysis: M.G., T.F., J.W., Å.M.; Funding and Resources: Å.M.; Writing – original and revised draft preparation: M.G., J.W., Å.M. All authors have read and agreed to the published version of the manuscript.
Conflicts of Interest
The self-disinfecting sink was invented by Micropharmics AB (owned by Å.M.) and developed together with Tunerlux AB. The funder had no role in study design, data collection and interpretation, or the decision to submit the work for publication.
References
- Coppola, N.; Maraolo, A.E.; Onorato, L.; Scotto, R.; Calo, F.; Atripaldi, L.; Borrelli, A.; Corcione, A.; De Cristofaro, M.G.; Durante-Mangoni, E.; Filippelli, A.; Franci, G.; Galdo, M.; Guglielmi, G.; Pagliano, P.; Perrella, A.; Piazza, O.; Picardi, M.; Punzi, R.; Trama, U.; Gentile, I. Epidemiology, Mechanisms of Resistance and Treatment Algorithm for Infections Due to Carbapenem-Resistant Gram-Negative Bacteria: An Expert Panel Opinion. Antibiotics (Basel). 2022, 11. doi: 10.3390/antibiotics11091263.
- Wagner, T.M.; Howden, B.P.; Sundsfjord, A.; Hegstad, K. Transiently silent acquired antimicrobial resistance: an emerging challenge in susceptibility testing. J Antimicrob Chemother. 2023, 78, 586-98. doi: 10.1093/jac/dkad024.
- Serra-Burriel, M.; Keys, M.; Campillo-Artero, C.; Agodi, A.; Barchitta, M.; Gikas, A.; Palos, C.; Lopez-Casasnovas, G. Impact of multi-drug resistant bacteria on economic and clinical outcomes of healthcare-associated infections in adults: Systematic review and meta-analysis. PLoS One. 2020, 15, e0227139. doi: 10.1371/journal.pone.0227139.
- Antimicrobial Resistance C. Global burden of bacterial antimicrobial resistance in 2019: a systematic analysis. Lancet. 2022, 399, 629-55. doi: 10.1016/S0140-6736(21)02724-0.
- MacVane, S.H. Antimicrobial Resistance in the Intensive Care Unit: A Focus on Gram-Negative Bacterial Infections. J Intensive Care Med. 2017, 32, 25-37. doi: 10.1177/0885066615619895.
- Sader, H.S.; Mendes, R.E.; Streit, J.M.; Carvalhaes, C.G.; Castanheira, M. Antimicrobial susceptibility of Gram-negative bacteria from intensive care unit and non-intensive care unit patients from United States hospitals (2018-2020). Diagn Microbiol Infect Dis. 2022, 102, 115557. doi: 10.1016/j.diagmicrobio.2021.115557.
- Decker, B.K.; Palmore, T.N. The role of water in healthcare-associated infections. Curr Opin Infect Dis. 2013, 26, 345-51. doi: 10.1097/QCO.0b013e3283630adf.
- Ladhani, H.A.; Yowler, C.J.; Claridge, J.A. Burn Wound Colonization, Infection, and Sepsis. Surg Infect (Larchmt). 2021, 22, 44-8. doi: 10.1089/sur.2020.346.
- Yin, Z.; Beiwen, W.; Zhenzhu, M.; Erzhen, C.; Qin, Z.; Yi, D. Characteristics of bloodstream infection and initial antibiotic use in critically ill burn patients and their impact on patient prognosis. Sci Rep. 2022, 12, 20105. doi: 10.1038/s41598-022-24492-z.
- Haider, M.H.; McHugh, T.D.; Roulston, K.; Arruda, L.B.; Sadouki, Z.; Riaz, S. Detection of carbapenemases bla(OXA48)-bla(KPC)-bla(NDM)-bla(VIM) and extended-spectrum-beta-lactamase bla(OXA1)-bla(SHV)-bla(TEM) genes in Gram-negative bacterial isolates from ICU burns patients. Ann Clin Microbiol Antimicrob. 2022, 21, 18. doi: 10.1186/s12941-022-00510-w.
- Lima, W.G.; Silva Alves, G.C.; Sanches, C.; Antunes Fernandes, S.O.; de Paiva, M.C. Carbapenem-resistant Acinetobacter baumannii in patients with burn injury: A systematic review and meta-analysis. Burns. 2019, 45, 1495-508. doi: 10.1016/j.burns.2019.07.006.
- Kotay, S.; Chai, W.; Guilford, W.; Barry, K.; Mathers, A.J. Spread from the Sink to the Patient: In Situ Study Using Green Fluorescent Protein (GFP)-Expressing Escherichia coli To Model Bacterial Dispersion from Hand-Washing Sink-Trap Reservoirs. Appl Environ Microbiol. 2017, 83. doi: 10.1128/AEM.03327-16.
- Valentin, A.S.; Santos, S.D.; Goube, F.; Gimenes, R.; Decalonne, M.; Mereghetti, L.; Daniau, C.; van der Mee-Marquet, N. A prospective multicentre surveillance study to investigate the risk associated with contaminated sinks in the intensive care unit. Clin Microbiol Infect. 2021, 27, 1347 e9- e14. doi: 10.1016/j.cmi.2021.02.018.
- Kearney, A.; Boyle, M.A.; Curley, G.F.; Humphreys, H. Preventing infections caused by carbapenemase-producing bacteria in the intensive care unit - Think about the sink. J Crit Care. 2021, 66, 52-9. doi: 10.1016/j.jcrc.2021.07.023.
- Loveday, H.P.; Wilson, J.A.; Kerr, K.; Pitchers, R.; Walker, J.T.; Browne, J. Association between healthcare water systems and Pseudomonas aeruginosa infections: a rapid systematic review. J Hosp Infect. 2014, 86, 7-15. doi: 10.1016/j.jhin.2013.09.010.
- Volling, C.; Ahangari, N.; Bartoszko, J.J.; Coleman, B.L.; Garcia-Jeldes, F.; Jamal, A.J.; Johnstone, J.; Kandel, C.; Kohler, P.; Maltezou, H.; Maze Dit Mieusement, L.; McKenzie, N.; Mertz, D.; Monod, A.; Saeed, S.; Shea, B.; Stuart, R.; Thomas, S.; Uleryk, E.; McGeer, A. Are Sink Drainage Systems a Reservoir for Hospital-Acquired Gammaproteobacteria Colonization and Infection? A Systematic Review. Open Forum Infect Dis. 2021, 8, ofaa590. doi: 10.1093/ofid/ofaa590.
- Bayuo, J.; Agbenorku, P. Coping strategies among nurses in the Burn Intensive Care Unit: A qualitative study. Burns Open. 2018, 2, 47-52. doi: 10.1016/J.BURNSO.2017.10.004.
- Schurch, A.C.; Arredondo-Alonso, S.; Willems, R.J.L.; Goering, R.V. Whole genome sequencing options for bacterial strain typing and epidemiologic analysis based on single nucleotide polymorphism versus gene-by-gene-based approaches. Clin Microbiol Infect. 2018, 24, 350-4. doi: 10.1016/j.cmi.2017.12.016.
- Pightling, A.W.; Pettengill, J.B.; Luo, Y.; Baugher, J.D.; Rand, H.; Strain, E. Interpreting Whole-Genome Sequence Analyses of Foodborne Bacteria for Regulatory Applications and Outbreak Investigations. Front Microbiol. 2018, 9, 1482. doi: 10.3389/fmicb.2018.01482.
- Sutterlin, S.; Dahlo, M.; Tellgren-Roth, C.; Schaal, W.; Melhus, A. High frequency of silver resistance genes in invasive isolates of Enterobacter and Klebsiella species. J Hosp Infect. 2017, 96, 256-61. doi: 10.1016/j.jhin.2017.04.017.
- Gideskog, M.; Welander, J.; Melhus, A. Cluster of S. maltophilia among patients with respiratory tract infections at an intensive care unit. Infect Prev Pract. 2020, 2, 100097. doi: 10.1016/j.infpip.2020.100097.
- Jung, J.; Choi, H.S.; Lee, J.Y.; Ryu, S.H.; Kim, S.K.; Hong, M.J.; Kwak, S.H.; Kim, H.J.; Lee, M.S.; Sung, H.; Kim, M.N.; Kim, S.H. Outbreak of carbapenemase-producing Enterobacteriaceae associated with a contaminated water dispenser and sink drains in the cardiology units of a Korean hospital. J Hosp Infect. 2020,104, 476-83. doi: 10.1016/j.jhin.2019.11.015.
- Catho, G.; Martischang, R.; Boroli, F.; Chraiti, M.N.; Martin, Y.; Koyluk Tomsuk, Z.; Renzi, G.; Schrenzel, J.; Pugin, J.; Nordmann, P.; Blanc, D.S.; Harbarth, S. Outbreak of Pseudomonas aeruginosa producing VIM carbapenemase in an intensive care unit and its termination by implementation of waterless patient care. Crit Care. 2021, 25, 301. doi: 10.1186/s13054-021-03726-y.
- Stjarne Aspelund, A.; Sjostrom, K.; Olsson Liljequist, B.; Morgelin, M.; Melander, E.; Pahlman, L.I. Acetic acid as a decontamination method for sink drains in a nosocomial outbreak of metallo-beta-lactamase-producing Pseudomonas aeruginosa. J Hosp Infect. 2016, 94, 13-20. doi: 10.1016/j.jhin.2016.05.009.
- Rehou, S.; Rotman, S.; Avaness, M.; Salt, N.; Jeschke, M.G.; Shahrokhi, S. Outbreak of carbapenemase-producing Enterobacteriaceae in a regional burn centre. J Burn Care Res. 2022, 43, 1203-6. doi: 10.1093/jbcr/irac067.
- Salm, F.; Deja, M.; Gastmeier, P.; Kola, A.; Hansen, S.; Behnke, M.; Gruhl, D.; Leistner, R. Prolonged outbreak of clonal MDR Pseudomonas aeruginosa on an intensive care unit: contaminated sinks and contamination of ultra-filtrate bags as possible route of transmission? Antimicrob Resist Infect Control. 2016, 5, 53. doi: 10.1186/s13756-016-0157-9.
- de Jonge, E.; de Boer, M.G.J.; van Essen, E.H.R.; Dogterom-Ballering, H.C.M.; Veldkamp, K.E. Effects of a disinfection device on colonization of sink drains and patients during a prolonged outbreak of multidrug-resistant Pseudomonas aeruginosa in an intensive care unit. J Hosp Infect. 2019, 102, 70-4. doi: 10.1016/j.jhin.2019.01.003.
- Fusch, C.; Pogorzelski, D.; Main, C.; Meyer, C.L.; El Helou, S.; Mertz, D. Self-disinfecting sink drains reduce the Pseudomonas aeruginosa bioburden in a neonatal intensive care unit. Acta Paediatr. 2015, 104, e344-9. doi: 10.1111/apa.13005.
- Diamond, F. Hot water might disinfect sinks better than chlorine. Infection control today. Available online: https://www.infectioncontroltoday.com/view/hot-water-might-disinfect-sinks-better-than-chlorine (accessed on 15 November 2022).
- Hota, S.; Hirji, Z.; Stockton, K.; Lemieux, C.; Dedier, H.; Wolfaardt, G.; Gardam, M. Outbreak of multidrug-resistant Pseudomonas aeruginosa colonization and infection secondary to imperfect intensive care unit room design. Infect Control Hosp Epidemiol. 2009, 30, 25-33. doi: 10.1086/592700.
- Longtin, Y.; Troillet, N.; Touveneau, S.; Boillat, N.; Rimensberger, P.; Dharan, S.; Gervaix, A.; Pittet, D.; Harbarth, S. Pseudomonas aeruginosa outbreak in a pediatric intensive care unit linked to a humanitarian organization residential center. Pediatr Infect Dis J. 2010, 29, 233-7. doi: 10.1097/INF.0b013e3181bc24fb.
- Alonso, C.A.; Gonzalez-Barrio, D.; Ruiz-Fons, F.; Ruiz-Ripa, L.; Torres, C. High frequency of B2 phylogroup among non-clonally related fecal Escherichia coli isolates from wild boars, including the lineage ST131. FEMS Microbiol Ecol. 2017, 93. doi: 10.1093/femsec/fix016.
Round 2
Reviewer 1 Report
The authors responded to all my queries and questions and the manuscript was significantly improved is now suitable to be published
Author Response
The reviewer is content, why no further responses are necessary from the authors.
Reviewer 2 Report
I accept the reply to the second point. However, the reply to the first point is still lacking. The delay in reporting the outbreak in the past can be taken into account to some extent, but I think the past cases should be able to give some implications for the current situation. Therefore, it is necessary to discuss this.
Author Response
Reviewer’s comment to our response: I accept the reply to the second point. However, the reply to the first point is still lacking. The delay in reporting the outbreak in the past can be taken into account to some extent, but I think the past cases should be able to give some implications for the current situation. Therefore, it is necessary to discuss this.
Authors’ response: Outbreaks with VIM-2-producing P. aeruginosa are extremely rare in the Nordic countries. As a matter of fact, the only two outbreaks that have taken place were interconnected and are described or referred to in our manuscript. With this manuscript the sequence of events becomes more complete and covers a period of 24 years, a rarity in itself. Although the outbreaks occurred and were declared over a long time ago, the causing clone has not left Lund University Hospital. If the treatment with acidic acid is discontinued, the clone grows in the sink samples. Obviously it is easier to stop an outbreak than to eradicate a bacterium from the plumbing system once it is established. This does give implications for both the current situation and the future. Outbreaks are spectacular and cost a lot, but they do not cost as much as the constantly present healthcare-associated infections (HAI). These everyday infections are also more difficult to discover, unless the bacteria carry some marker in the form of resistance genes. A British research group used whole-genome sequencing some years ago, to show that “environmental bacterial populations are largely structured by ward and sink, with only a handful of lineages being widely distributed. suggesting different prevailing ecologies, which may vary as a result of different inputs and selection pressures" (Constantinides B, Chau KK, Quan TP, Rodger G, Andersson MI, Jeffery K, Lipworth S, Gweon HS, Peniket A, Pike G, Millo J, Byukusenge M, Holdaway M, Gibbons C, Mathers AJ, Crook DW, Peto TEA, Walker AS, Stoesser N. Genomic surveillance of Escherichia coli and Klebsiella spp. in hospital sink drains and patients. Microb Genom. 2020 Jul;6(7):mgen000391. doi: 10.1099/mgen.0.000391). They also compared the environmental isolates with contemporaneous patient isolates, and suggested that sinks may contribute to up to 10% of the E. coli-induced infections. In our own study exploring the efficiency of a self-disinfecting (recently submitted to Microorganisms), the transmission rate from sink to patient was 12.5%. The yearly cost for HAIs in Sweden is 150-220 million EUR. About 40% of these infections are urinary tract infections, and a clear majority of them is caused by E. coli. If translated into money, transmission from sinks could cost the Swedish health care several million EUR per year, not to mention all the unnecessary suffering and deaths. A paragraph about this has been added to the Discussion.
